# Canonical WNT signaling-dependent gating of *MYC* requires a noncanonical CTCF function at a distal binding site

Ilyas Chachoua[1,3], Ilias Tzelepis [1,3], Hao Dai[1,2,3], Jia Pei Lim[1,3], Anna Lewandowska-Ronnegren[1,3], Felipe Beccaria Casagrande [1], Shuangyang Wu[1], Johanna Vestlund [1], Carolina Diettrich Mallet de Lima[1], Deeksha Bhartiya[1], Barbara A. Scholz[1], Mirco Martino [1], Rashid Mehmood [1] & Anita Göndör [1✉]

Abnormal WNT signaling increases *MYC* expression in colon cancer cells in part via oncogenic super-enhancer-(OSE)-mediated gating of the active *MYC* to the nuclear pore in a poorly understood process. We show here that the principal tenet of the WNT-regulated *MYC* gating, facilitating nuclear export of the *MYC* mRNA, is regulated by a CTCF binding site (CTCFBS) within the OSE to confer growth advantage in HCT-116 cells. To achieve this, the CTCFBS directs the WNT-dependent trafficking of the OSE to the nuclear pore from intranucleoplasmic positions in a stepwise manner. Once the OSE reaches a peripheral position, which is triggered by a CTCFBS-mediated *CCAT1* eRNA activation, its final stretch (≤0.7 μm) to the nuclear pore requires the recruitment of AHCTF1, a key nucleoporin, to the CTCFBS. Thus, a WNT/ß-catenin-AHCTF1-CTCF-eRNA circuit enables the OSE to promote pathological cell growth by coordinating the trafficking of the active *MYC* gene within the 3D nuclear architecture.

[1] Department of Oncology and Pathology, Bioclinicum, Karolinska University Hospital, U2, Akademiska Stråket 1, Karolinska Institutet, Stockholm, Sweden. [2] Present address: Department of Breast Disease, Henan Breast Cancer Center, The affiliated Cancer Hospital of Zhengzhou University & Henan Cancer Hospital, Zhengzhou, China. [3] These authors contributed equally: Ilyas Chachoua, Ilias Tzelepis, Hao Dai, Jia Pei Lim, Anna Lewandowska-Ronnegren. ✉email: anita.gondor@ki.se

The original proposal of the gene-gating hypothesis more than 30 years ago[1] outlined the principle that the juxta-position of inducible genes to the nuclear pore would facilitate their reactivation and the rapid nuclear export of derived mRNAs upon repeated stimuli[2,3]. While such events have subsequently been well documented in both yeast[4] and flies[5], they have been much more elusive in mammalian cells. This situation likely reflects that the nuclear volumes of mammalian cells are vastly larger[6] to compound direct comparisons with yeast, for example. We have earlier provided indirect evidence that the colorectal super-enhancer recruited the transcriptionally active MYC alleles to nuclear pores from intra-nucleoplasmic positions to increase the rate of nuclear export of MYC mRNAs[7]. This process increased the overall MYC expression level by enabling the exported mRNA to escape from the more rapid decay kinetics in the nucleus compared to the cytoplasm[7]. Importantly, the facilitated nuclear export rate of MYC mRNAs was the sole parameter responsible for the increased MYC expression in colon cancer cells (HCT-116) compared to primary human colon epithelial cells. Although the suggested link of this process to the distal (>500 kb) OSE[7] is congruent with a version of the gene-gating phenomenon also in human cancer cells, the underlying mechanisms are not known. To deal with these shortcomings, we focus here on the potential regulatory cis elements within the colorectal OSE. Among the candidates, the CTCFBS, positioned within an eRNA gene, CCAT1, stood out for a number of reasons. First, CTCF has been described as a master regulator of the genome, attributed to a range of pivotal processes[8]. These include the organization of chromatin insulators, boundaries[9], enhancer–gene interactions via the cohesin complex[10], as well as the mediation of the rhythmic recruitment of active circadian genes to the lamina for subsequent repression[11]. Second, it binds to a region within the OSE that shows physical proximity with MYC in colon cancer cells[7], and third, CTCF is linked with long-range regulation of MYC expression[12,13]. Specifically, it has been argued that OSE-MYC interactions are facilitated by the CCAT1 eRNA when complexed with CTCF[14,15].

To assess if this CCAT1-specific CTCFBS plays a functional role in the gene-gating process, we have edited 8 bps within its site following a previously used strategy[16–19], by using the CRISPR technology. The comparison of two cell clones (D3 and E4) carrying the mutated OSE allele with the wild type (WT) parental HCT-116 cells provides genetic evidence that the CTCFBS is necessary for the WNT-regulated increase in the nuclear export rate of MYC mRNAs. This CTCF function reflects, moreover, its ability to direct the trafficking of the active MYC allele to the nuclear periphery in a stepwise manner. Thus, it is responsible for the WNT-dependent activation of CCAT1 eRNA that we could link to the repositioning of MYC from the nuclear interior to a location juxtaposed to the periphery. The next step, promoting MYC to migrate from this position to the nuclear periphery/pore (encompassing <1 μm), is triggered by the WNT-regulated interaction between CTCF and AHCTF1—a key nucleoporin essential for the gating process[7]—potentially facilitated by ß-catenin. Thus, the mutant OSE allele is, in contrast to the WT allele, unable to efficiently recruit AHCTF1, resulting in a significantly reduced representation of both the OSE and MYC regions at nuclear periphery/pore-proximal positions in mutant cells. However, the lack of a functional CTCFBS does not affect the ability of the OSE to physically interact with MYC, documenting a function of the OSE-specific CTCFBS, which is independent of the well-known ability of CTCF to connect distant chromatin fibers[9,17,18]. Importantly, the WT HCT-116 cells display a proliferative advantage over the cells carrying the edited OSE alleles due to reduced MYC expression levels in the mutant cells. Since the gating process is specific for colon cancer cells[7],

these findings open up potential strategies to both diagnose cancer cells at risk of developing pathological gene-gating and antagonize pathological MYC expression without affecting the normal MYC function. Finally, the designation of the gating function to the versatile CTCF[20] raises the question whether this feature is not restricted to the OSE, but applies genome wide.

## Results

**CTCF binding to an oncogenic super-enhancer distal to MYC confers an excessive growth advantage to colon cancer cells.** While CTCF binding to a site within the OSE-specific eRNA gene (CCAT1) is prominent in HCT-116 cells, it was absent from the corresponding region in normal colon epithelial cells (Fig. 1a, b). Since the normal cell counterparts (human colon epithelial cells, HCECs) lack functional MYC gating[7], this correlation encouraged us to target this particular CTCFBS using the CRISPR technique to generate HCT-116 cells with edited OSE alleles that no longer bind CTCF. We chose to target the C/G sites within the CTCFBS (the edited sequences are marked in gray in Fig. 1a), similar to a strategy that successfully identified in vivo a chromatin insulator in the mouse[17,21]. Two clones (D3 and E4) were generated, with identical mutations within the CCAT1-specific CTCFBS in all alleles (Fig. 1c). To assess the genome-wide off-target effects of CRISPR–Cas9 editing, we sequenced the genomes of the WT, D3 and E4 cells and adapted a genome wide off-target detection pipeline modified from GOTI[22] (Supplementary Fig. 1a). Allowing for a filter of 100% sequence similarity for a blast word size ≥12 nucleotides, we found no sgRNA-associated target for the D3 and E4 cell clones in the blast results. We further analyzed the potential off-targets using Cas-OFFinder (see Methods) allowing for 5 mismatches in the settings to directly align the sgRNA with the genome. No overlap was found between the variant location and potential off-target sites to indicate that the editing process did not as such generate genome wide variants. However, we did find a total of 91 indels and 10 SNVs that were shared between the D3 and E4 cell clones. Two of the indels that were common to both cell clones mapped to the vicinity of CTCFBSs on chromosome 8 and 22. ChIP-seq analyses showed that these changes did not antagonize CTCF binding (Supplementary Fig. 1b). Since neither of these regions interacted with MYC or the OSE, they are unlikely to be involved in the MYC gating process.

We next examined if the edited CTCFBS retained an ability to interact with CTCF. ChIP assays showed that the mutated OSE allele lost >90% of CTCF binding compared to the WT allele in the parental HCT-116 cell population (Fig. 1b). In contrast, the level of CTCF binding to internal controls, such as the MYC promoter and the H19 imprinting control region, was statistically indistinguishable between the WT and mutant HCT-116 cells (Fig. 1b). To generate an overview of the CTCF binding patterns in the OSE-MYC region, the WT cells and D3/E4 cell clones were subjected to ChIP-seq analyses. The patterns, resulting from normalization of three independent ChIP-seq experiments, showed that, apart from the CCAT1-specific CTCFBS, all other CTCFBS within the OSE-MYC domain bound CTCF equally well in WT and D3/E4 cells (Fig. 1d).

To assess if the mutation of the CTCFBS translated into a phenotypic trait, we developed primers and PCR conditions to determine the allele-specific representation of the WT and mutant (D3/E4) OSE alleles (Supplementary Fig. 1c) in co-cultures that were maintained for up to 2 weeks. We chose this strategy over the analyses of growth curves for the individual cell populations, as it provided an internal control directly enabling a comparison of the proliferation rate of WT and mutant cells. By scoring for the ratios between WT and D3 or E4 OSE alleles we

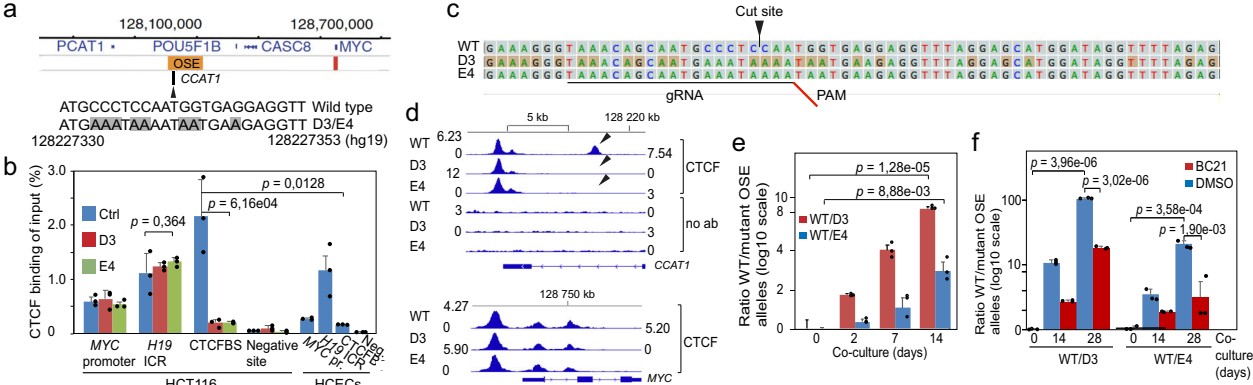

**Fig. 1 The CTCFBS within the OSE-specific eRNA gene (*CCAT1*) confers a proliferative advantage to the HCT-116 cells. a** The position of the *CCAT1*-specific CTCFBS within the OSE is indicated (black arrow). The core binding sequence was modified at 8 bases, as marked by gray boxes in the panel, by CRISPR editing. Orange boxes depict enhancer regions. **b** ChIP analyses of the occupancy of the CTCFBS within the OSE. The *MYC* promoter and the *H19* ICR were used as positive controls. Neg. CTCF negative site (see Methods). **c** DNA sequences in the edited CTCFBS in comparison to WT HCT-116 cells. **d** ChIP-seq profiles of CTCF binding patterns to a region encompassing the OSE (upper) and *MYC* (lower) regions. The ChIP-seq data, which is visualized in relation to a genome browser snapshot, was normalized from three independent experiments for WT HCT-116, D3, and E4 cells. The boxed motif, representing the *CCAT1* gene region, is enlarged to identify the edited CTCFBS within its intron. The arrows identify the edited *CCAT1*-specific CTCFBS. **e** Co-cultures of wild type and mutant HCT-116 cells harvested at the indicated time points, followed by qPCR analyses of the proportion of WT and mutant CTCFBSs, respectively. **f** The effect of BC21 on the relative growth rate of the WT, D3, and E4 cells. All the genomic coordinates use hg19 as a reference genome. The data represent in all instances the average of three independent experiments with indicated standard deviation. The *p* values were calculated by the two-tailed Student's *t* test.

could determine that the WT *CCAT1*-specific CTCFBS confers excessive growth advantage to the WT cells that out-grew the D3/E4 clones already within a week (Fig. 1e). Given the link between pathological WNT signaling and *MYC* gating[7], we next addressed if the inhibition of the canonical ß-catenin function would specifically target this growth advantage. Since ß-catenin is mutated in HCT-116 cells[23], we aimed at antagonizing its function in an as direct manner as possible. To this end, we used the BC21 drug, which we have earlier shown to specifically antagonize the proximity between ß-catenin and TCF4 as well as to evict ß-catenin from the OSE chromatin proximal to the CTCFBS[7]. The BC21 drug was optimized to antagonize the physical interaction between ß-catenin and TCF4 (Supplementary Fig. 1d) without interfering with *MYC* transcription (see below). Figure 1f shows that BC21 treatment indeed reduced the growth advantage of the WT HCT-116 cells in both the WT/D3 and WT/E4 co-cultures ca 6-fold during a 2-week co-culturing. We therefore conclude that the binding of CTCF to the intron of the *CCAT1* gene endows colon cancer cells with a WNT signaling-dependent growth advantage that efficiently outcompetes a cell population lacking this functional CTCFBS.

**The *CCAT1*-specific CTCFBS within the oncogenic super-enhancer controls the nuclear export rate of the mRNAs produced from the interacting, distal *MYC*, and *FAM49B* genes.** We have earlier shown that the OSE controls *MYC* expression post-transcriptionally in a WNT signaling-dependent manner in HCT-116 cells[7]. To assess the potential involvement of the *CCAT1*-specific CTCFBS in this process, we first addressed if the facilitated rate of nuclear export of *MYC* mRNAs, a hallmark of the mammalian gating principle[7], was impaired in the mutant D3 and E4 cells in comparison to the WT HCT-116 cells. This assay was performed by determining the cytoplasmic/nuclear ratio of newly synthesized mRNAs over time, as described earlier[7]. We explored the nuclear export rate of both the *MYC* and *FAM49B* mRNAs, since the OSE physically interacts also with an enhancer proximal to *FAM49B*, as determined by Nodewalk analyses[24], and because the *MYC* and *FAM49B* gene products are functionally

intertwined[25,26]. Figure 2a shows that both mutant cell clones displayed more than threefold reduction in the rate of nuclear export of *MYC* and *FAM49B* mRNAs, without displaying any significant change in the overall transcriptional rate of either *MYC* or *FAM49B* (Fig. 2b). Moreover, the nuclear export rate in the mutant cells was, in contrast to the parental WT HCT-116 cells, unaffected when treated with BC21 (Fig. 2a). Of note, the level of *MYC* transcription per cell was unaffected by the editing of the *CCAT1*-specific CTCFBS (Fig. 2b). The presence of two *MYC/FAM49B* alleles per mutant cell as opposed to three copies in the WT cells (see Source data) indicated that *MYC/FAM49B* transcription per allele might be higher in the mutant cells (see also Supplementary Fig. 2a, b). Although this observation is not statistical significant, it suggests that the OSE-specific CTCFBS does not reduce *MYC* and *FAM49B* transcription per se. As could be expected, the reduced rate of nuclear export of *MYC* and *FAM49B* mRNAs in the mutant cells correlated with significantly reduced total *MYC* and *FAM49B* mRNA expression levels, which were similar for both D3 and E4 clones, in comparison with WT HCT-116 cells (Fig. 2c). We next modeled the observed reduction of *MYC* expression in the mutant cells using the parameters of transcription, mRNA decay and export rates, as previously described[7]. Figure 2d demonstrates that, based on these parameters, the simulated *MYC* expression difference between the WT HCT-116 cells and mutant D3 or E4 clones agreed with the experimental data. We thus conclude that the *CCAT1*-specific CTCFBS increases *MYC* expression solely by facilitating the nuclear export rate of its derived mRNA - a feature that provides the WT HCT-116 cells with a strong proliferative advantage.

**ß-catenin and CTCF collaborate in recruiting the nuclear pore-anchor AHCTF1 to the oncogenic super-enhancer.** Given that AHCTF1 is both a key factor in the anchoring of the OSE to nuclear pores and essential for the trafficking of *MYC* in colon cancer cells[7], we next explored potential relationships between CTCF and AHCTF1 using co-immunoprecipitation analyses. Remarkably, the results showed that the recovery of AHCTF1 in these samples generally exceeded that of CTCF (Fig. 3a, b).

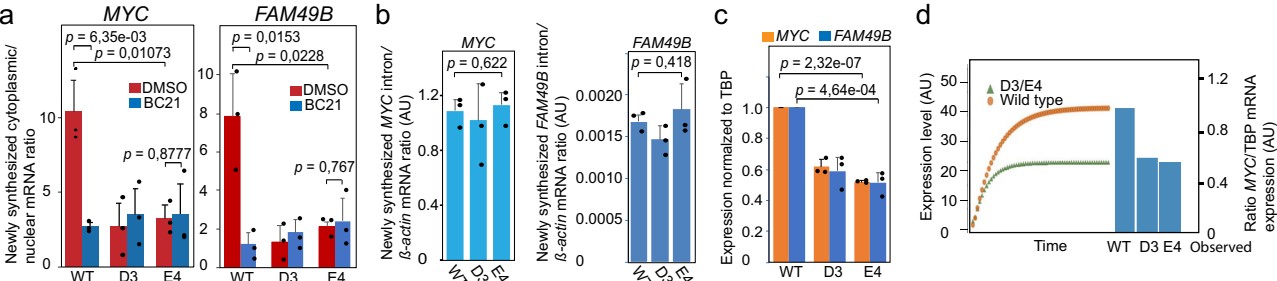

**Fig. 2 The *CCAT1*-specific CTCFBS increases *MYC* and *FAM49B* expression by facilitating the nuclear export of *MYC* and *FAM49B* mRNAs in a WNT-dependent manner. a** The rate of nuclear export of newly synthesized *MYC* and *FAM49B* mRNAs in WT and mutant HCT-116 cells in the absence or presence of BC21 (see Methods). **b** The transcriptional rate of the *MYC* and *FAM49B* genes. The data were generated by qRT-PCR analyses of *MYC/FAM49B* transcription using newly synthesized (30 min ethynyl-uridine pulse) RNA as template and normalized to *ACTB* transcription. **c** The steady state levels of cytoplasmic *MYC* and *FAM49B* mRNAs in WT and mutant HCT-116 cells normalized to TBP expression and external "spike in" RNA controls. The levels of TBP and ß-actin mRNA expression, markers used to normalize the mRNA export and transcription rates, were not significantly different between the WT and D3/E4 cells and correctly estimated the number of input cells (Supplementary Fig. 2a, b). **d** Comparison between observed and simulated cytoplasmic *MYC* RNA levels in WT HCT-116 and E4 cells. The simulation used the parameters of nuclear export of *MYC* mRNA (**a**), transcriptional rate (**b**), and the kinetics of *MYC* mRNA decay in the nuclear and cytoplasmic compartments, as described before[7]. The data represent in all instances the average of three independent experiments with indicated standard deviation. The *p* values were calculated by the two-tailed Student's *t* test.

This observation strongly indicated that a subpopulation of CTCF physically interacts with oligomers of AHCTF1. Of note, the expression levels (Supplementary Fig. 3a, b) and complex formation between CTCF and AHCTF1 by co-immunoprecipitation analyses (Supplementary Fig. 3c) were very similar between WT and D3/E4 cells to rule out clonal variation as an underlying cause of the effects of the edited CTCFBS. Since AHCTF1 is part of the nuclear pore as 16-mers[27], a fraction of CTCF might interact directly or indirectly with the nuclear pore and/or a pre-nucleopore complex. In line with this reasoning, the relatively high frequency of interactions between CTCF and NUP133 might thus reflect that a pre-nucleopore complex[28] interacts with CTCF via AHCTF1. Our earlier observation that BC21 evicted not only ß-catenin but also AHCTF1 and NUP133 from the OSE chromatin[7] indicated that ß-catenin is also involved in the CTCF-AHCTF1 complex formation. We therefore examined if the physical link between CTCF and AHCTF1 could be sensitive to BC21 treatment, again using co-immunoprecipitation analyses. Figure 3c shows that BC21 indeed counteracted the formation of the CTCF-AHCTF1 complex. Moreover, as BC21 evicted AHCTF1 from the OSE chromatin (Fig. 3d) without negatively affecting CTCF binding to the CTCFBS (Supplementary Fig. 3d), ß-catenin might facilitate the recruitment of AHCF1 to the *CCAT1*-specific CTCFBS. To scrutinize the role of CTCFBS-bound CTCF in more detail, we examined if a reduction in CTCF expression levels by siRNA treatment would affect the presence of AHCTF1 at this CTCFBS. Figure 3e shows that reduced expression of CTCF (Supplementary Fig. 3e) impaired the binding of both CTCF and AHCTF1 in WT HCT-116 cells to the *CCAT1*-specific CTCFBS. Since the mutated CTCFBS also displayed a reduced ability to bind AHCTF1 in both the D3 and E4 clones (Fig. 3f), we conclude that the CTCF-CTCFBS complex likely collaborates with ß-catenin to promote the recruitment of AHCTF1 to the OSE.

To explore if AHCTF1 has any role in the distribution of the OSE within the nucleus, we performed DNA FISH analyses of formaldehyde-fixed WT HCT-116 cells transfected with siGFP or siAHCTF1. The DNA FISH signals scoring for the distances of the OSE alleles from the nuclear periphery are presented in a bar diagram with distance windows ranging from <0.3 to >1.5 µm, as described before[7]. The results show that a reduction in cellular AHCTF1 expression (Fig. 3g) impedes the ability of the OSE to

make the final stretch to the periphery/pore (Fig. 3h). The window of 0,7 micrometers from the periphery represented the highest statistical significance (Fig. 3i) in altered OSE distribution upon AHCTF1 downregulation. This data therefore suggests that the interaction between CTCF and AHCTF1 most likely occurs prior to the final anchoring of the OSE to the nuclear pore. Of note, although AHCTF1 is primarily localized at nuclear pores, a considerable fraction can also be found in the nucleoplasm[28]. Indeed, in situ proximity analyses (ISPLA)[11] showed that the highest potential for interactions between CTCF and AHCTF1 spanned a region 1-2 micrometers distal to the nuclear periphery (Fig. 3j, k and Supplementary Fig. 3f), similar to the CTCF-NUP133 ISPLA signals (Supplementary Figs. 3f, 4a, b). Strikingly, this potential for interaction was significantly reduced in WT cells treated with BC21 (Fig. 3j, k), reinforcing the notion that CTCF and the ß-catenin/TCF4 complex collaborate in complex formation with AHCTF1.

All of these observations converge on a model where CTCF and ß-catenin act as major players in recruiting AHCTF1 to the OSE and hence the completion of the gating of *MYC* to nuclear pores. Interestingly, both TCF4, which has binding motifs on either side of the *CCAT1*-specific CTCFBS (Supplementary Fig. 5a), and ß-catenin showed prominent binding to the OSE not only in WT HCT-116, but also in D3 and E4 cells (Supplementary Fig. 5b) that were shown above to display reduced AHCTF1 binding to the CTCFBS. Taken together with that: (i) BC21 evicts AHCTF1 (Fig. 3d) and ß-catenin[7], but not CTCF, from the OSE (Supplementary Fig. 3d), (ii) BC21 impairs the physical interaction between CTCF-AHCTF1 as well as the proximities between CTCF and AHCTF1 when juxtaposed to the periphery (Fig. 3c, j, k), and (iii) knock-down of CTCF expression reduces the presence of AHCTF1 at the OSE (Fig. 3e), we submit that CTCF and ß-catenin join forces in promoting the presence of AHCTF1 at the OSE. We argue, moreover, that despite the absence of a strong interaction between ß-catenin and CTCF in HCT-116 cells (Fig. 3a, b), these factors collaborate in recruiting and/or stabilizing the presence of AHCTF1 at the OSE with AHCTF1 likely functioning as a bridge between these factors (Fig. 4). According to this scenario, the hypothesized accumulation of AHCTF1 at the CTCFBS as oligomers might enable its lateral distribution to the *CCAT1* promoter within the OSE (Fig. 3d, f).

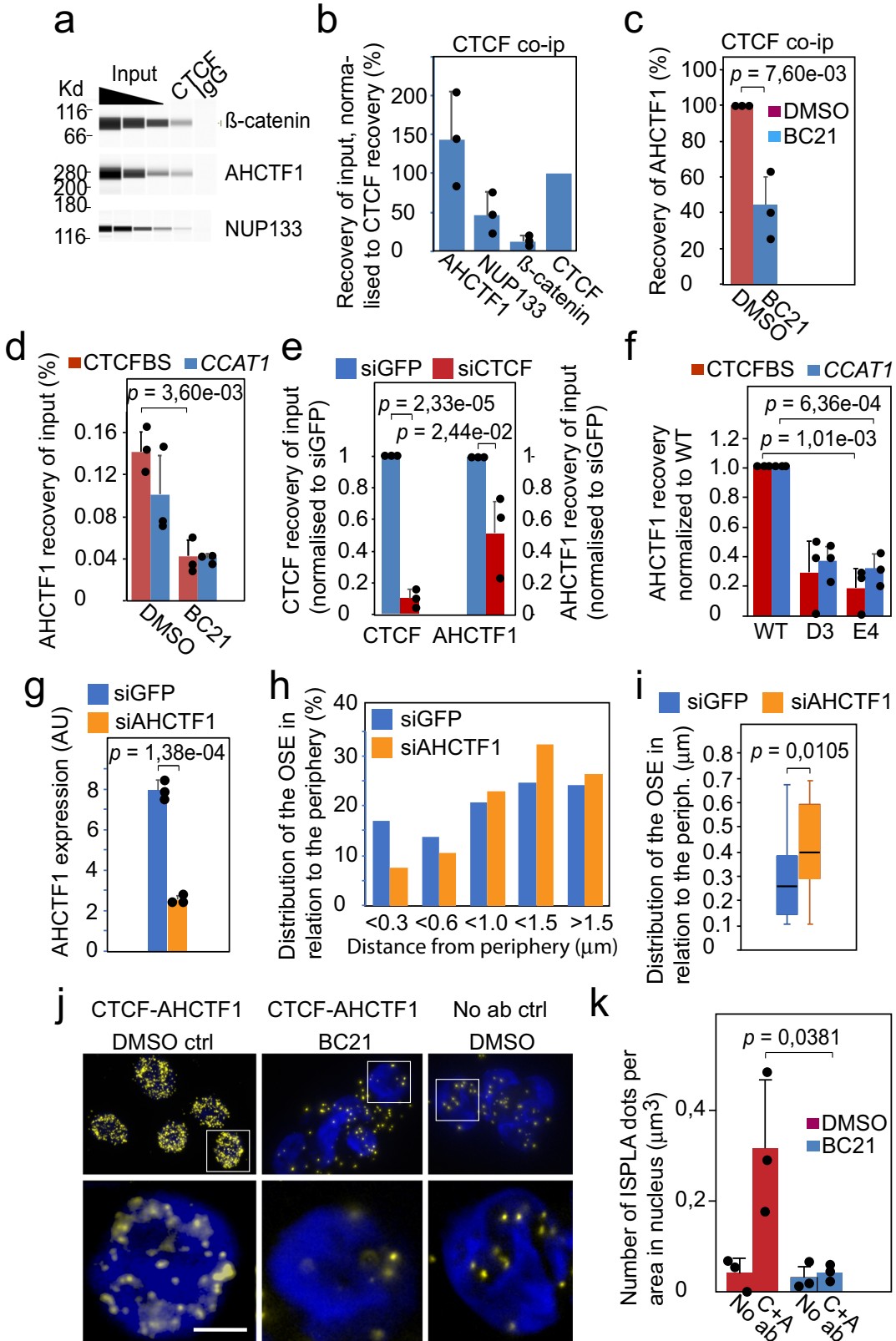

**CTCF coordinates the proximity between the OSE and *MYC* regions at the nuclear periphery, but is not directly responsible for their interactions**. To explore how the gating process is coordinated with interactions between the OSE-*MYC* regions, we first addressed their distributions within the nuclear architecture using 3D DNA FISH probes covering 8 and 9 kb of the OSE and

*MYC* regions, respectively (Fig. 5a)[7]. To visualize the relationships between the generated FISH signals, we calculated the value "c" from the equation c = b − a, where "b" and "a" are the distances in micrometers between the nuclear periphery and the *MYC* or the OSE signals, respectively[7]. The data was further stratified by including information about single-single

**Fig. 3 CTCF and ß-catenin recruit AHCTF1 to the oncogenic super-enhancer to promote its ability to reach the nuclear pore. a** Co-immunoprecipitation analyses of physical interactions between CTCF, ß-catenin, NUP133 and AHCTF1. IgG negative control. **b** Quantification of the CTCF-bound complexes shown in (**a**). **c** Co-immunoprecipitation analyses of the physical interactions between AHCTF1 and CTCF in WT HCT-116 cells in the absence or presence of BC21. **d** ChIP analyses of the binding of AHCTF1 to the oncogenic super-enhancer in DMSO control or BC21-treated WT HCT-116 cells. **e** ChIP analyses of CTCF and AHCTF1 binding to the *CCAT1*-specific CTCFBS in cells transfected with siGFP or siCTCF. The signals were normalized to the siGFP controls. The average siCTCF-mediated reduction in CTCF expression was 85% (Supplementary Fig 3d). **f** ChIP analyses of AHCTF1 binding to the CTCFBS and the *CCAT1* promoter within the OSE in WT HCT-116 and mutant (D3/E4) cells. **g** The knock-down of AHCTF1 expression by siRNA using a siGFP as control. **h** 3D DNA FISH analyses of the proximity between the OSE and the nuclear periphery in HCT-116 cells in the presence or absence of AHCTF1 (reduced to 72% in comparison to controls[7]). The bars represent the sum of two independent experiments (219 and 201 alleles, respectively) for siGFP and siAHCTF1-treated cells. **i** Box-and-whisker plots show median values, interquartile ranges and Tukey whiskers of the distribution of the OSE within 0.7 μm from the nuclear periphery. **j** In situ proximity ligation assay (ISPLA) of the proximity between CTCF and AHCTF1 in the absence or presence of BC21 in WT HCT-116 cells. Overviews of the DMSO, BC21, and no primary antibody control motifs (upper row), with the the lower row shows magnifications of focal planes marked in the upper row. Bar = 5 μm. **k** The quantification of the ISPLA signals. The data is based on three independent experiments counting a total number of 710 alleles. C CTCF antibody, A AHCTF1 antibody, No ab no primary antibody. All the data (except for **h**) represent the average of three independent experiments with indicated standard deviations. The *p* values for (**b**–**g**, **k**) were calculated by the two-tailed Student's *t* test whereas the *p* value for (**i**) was calculated using the two-sided KS test.

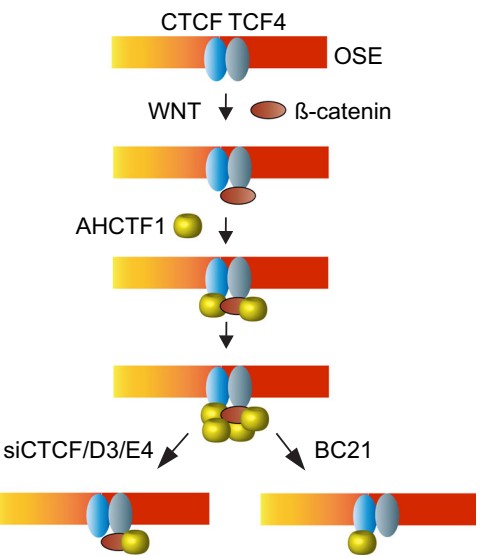

**Fig. 4 Schematic model of the recruitment of AHCTF1 to the OSE-specific CTCFBS.** Both WT HCT-116 and D3/E4 cell clones displayed prominent TCF4/ß-catenin binding to the *CCAT1*-specific CTCFBS region, independently of the mutation in the CTCFBS (Supplementary Fig. 5b). Since both the mutation of the CTCFBS (Fig. 3d, f) and the disruption of ß-catenin-TCF4 complex[7] lead to a reduction in AHCTF1 presence at the OSE, we propose that the juxtaposed CTCF and TCF4-binding sites (Supplementary Fig. 5a) collaborate to stabilize the presence of AHCTF1 at the *CCAT1*-specific CTCFBS via a CTCF-AHCTF1-ß-catenin complex. The timing of these events is largely unknown and are thus hypothetically visualized in the image.

(primarily G1), single-double (primarily early S phase) or double-double (primarily late S + G2) DNA FISH signals to explore any relationship to the cell cycle. Figure 5a, b shows that the OSE and *MYC* alleles approach the nuclear periphery in a coordinated manner (c ≈ 0 at the periphery), which is in keeping with the previously reported observation that they are generally closest to each other when approaching the nuclear periphery in WT HCT-116 cells[7]. The proximity of both the OSE and *MYC* regions to the periphery as well as the coordination of their recruitment to the nuclear periphery (as indicated by the "c" value) were, however, strongly reduced in both the E4 (Fig. 5a, b) and D3 (Supplementary Fig. 6) mutant cells. The statistical significance of the differential sub-nuclear localization of the OSE and *MYC* between WT and mutant HCT-116 cells

was determined by examining the cumulative distribution of the OSE and *MYC* alleles for the WT, D3 and E4 cell populations spanning one micrometer from the nuclear periphery. Of note, only the un-replicated alleles showed a significant effect on the proximity between the OSE and the nuclear periphery (Fig. 5c, d and Supplementary Fig. 6), suggesting that the gating principle is specific for the G1 phase of the cell cycle. We also observed that the presence of the *MYC* alleles at the periphery was significantly reduced in both D3 and E4 cells (Fig. 5c, d) to reinforce our earlier observation that it is the OSE region that brings *MYC* to the nuclear periphery/pore and not vice versa[7].

Although such data would seem to indicate that CTCF directly facilitates communications between the OSE and *MYC*[13,14], this is likely not the case. The reason for this conclusion is based on data generated by the Nodewalk technique[7,24], a 3C-like method, which analyses physical interactions between distal regions within the living cell in an ultra-sensitive manner. Using *MYC* as an anchor, the Nodewalk analyses generated a total of 22972 deduplicated reads or ligation events from a total input representing ca 60000 WT, D3 and E4 cells in total, respectively, from three independent experiments. When plotted in bar diagrams, showing the percentage of reads mapping to the OSE (Fig. 5e), or mapped on the physical map of the OSE-*MYC* region (Fig. 5f), it appears clear that the *CCAT1*-specific CTCFBS has no role in the overall interaction patterns between *MYC* and the OSE. This observation is not entirely surprising given the scarcity of CTCF binding sites within the OSE and the multiple contacts between the entire OSE and the *MYC* anchor. Indeed, the Mediator complex that covers the OSE[29] is known to mediate enhancer–gene interactions[30]. Nonetheless, the *CCAT1*-specific CTCFBS might indirectly promote physical interactions between the OSE and *MYC* regions by directing their recruitment to the crowded environment represented by the nuclear periphery[5].

**WNT-dependent activation of *CCAT1* transcription is mediated by CTCF to drive the peripheral positioning of the oncogenic super-enhancer.** Recent observations have implicated the *CCAT1* eRNA as a mediator of OSE-*MYC* interactions[14], potentially via RNA–RNA interactions[15]. The involvement of specific *CCAT1* eRNA isoforms in these processes, with different 3′-end (*CCAT1*-L) or 5′-ends (*CCAT1*-5L), complicates, however, interpretation (Fig. 6a). In Supplementary Fig. 7a we show that while the WT HCT-116 cells do not express the *CCAT1*-5L version, they prominently express the *CCAT1*-L version. To assess the potential ability of this transcript and/or the transcriptional activity of *CCAT1* to facilitate OSE-*MYC* proximity, we first performed RNA FISH analyses to determine

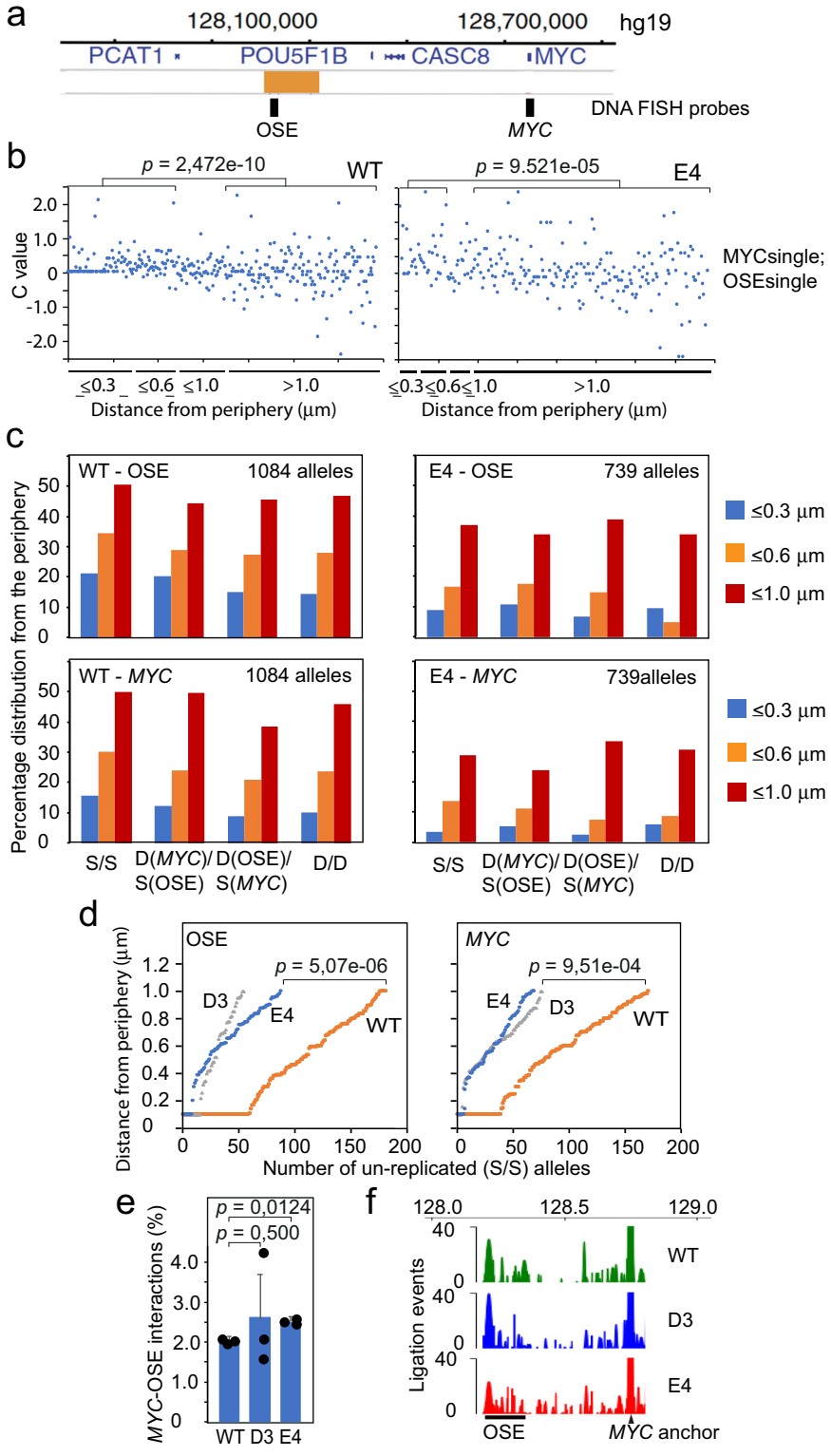

the distribution of active *CCAT1* alleles, followed by DNA FISH analyses using OSE- and *MYC*-specific probes to determine their proximities. As has been reported earlier[14,15], we observed a generally strong RNA FISH signal peaking at a nuclear periphery-proximal position that declined when its template approached very close the nuclear periphery/pore (Fig. 6b, c). In contrast to a previous claim[14], however, we were unable to tease out any relationship between OSE-*MYC* proximities and the position or strength of the *CCAT1* eRNA RNA FISH signal

(Supplementary Fig. 7b). This result is congruent with our observations that the OSE and *MYC* regions are most proximal to each other when positioned within 0.5 μm from the periphery[7] (Fig. 5a, b)—a sub-compartment with declining *CCAT1* eRNA transcription (Fig. 6b, c). Importantly, *CCAT1* alleles displaying little or no transcriptional activity were more distal to the nuclear periphery (marked in red, Fig. 6c) than alleles displaying higher transcription (marked in blue, Fig. 6c). This observation, which is in keeping with an increased mobility

**Fig. 5 The *CCAT1*-specific CTCFBS influences the proximity between the OSE and *MYC* at the nuclear periphery, but not their overall interaction frequency. a** Schematic map (to scale) of the OSE and *MYC* regions with the position of the DNA FISH probes indicated. **b** Analysis of the "c" value (scoring for the difference in the distances of *MYC* and the OSE from the nuclear periphery) in relation to the proximity of the OSE to the nuclear periphery in control and mutant HCT-116 cells for un-replicated alleles (MYCsingle/OSEsingle). The replication state of the *MYC* and OSE regions are indicated by the number of replicated alleles (see Supplementary Fig. 6 for additional information). A total of 1085 (Ctrl), and 740 (E4) alleles were counted from two independent experiments. **c** The overall proximity between the OSE and *MYC* regions from the nuclear periphery were stratified into three distances. S/S = un-replicated alleles; D(MYC)/S(OSE) = The *MYC* allele replicated before the OSE allele; S(MYC)/D(OSE) = The OSE allele replicated before the *MYC* allele; D(MYC)/D(OSE) = Both *MYC* and OSE alleles replicated. **d** The cumulative distribution of un-replicated *MYC* and OSE alleles within one micrometer from the nuclear periphery in WT HCT-116, D3, and E4 cells. The numbers have been derived from the source data of (**b**, **c**). **e** Chromatin fiber interaction analyses (Nodewalk)[7,24] showing the percentage of sequences within the OSE region (hg19:chr8:128192176-128309374) that interacted with the *MYC* anchor (hg19:chr8:128,746,000-128,756,177). **f** Graphic representation of the interaction patterns in the 5′-flank of the *MYC* anchor. The data is the average of unique ligation events normalized from three independent experiments. The *p* values indicated in panels b and d were determined using the two-sided KS test, while the *p* values in panel e were determined by the two-tailed Student's *t* test.

of transcriptionally active genes[31], is further documented in Fig. 6d. Thus, box plot analyses show a direct and significant correlation between the *CCAT1* eRNA RNA FISH signal and the distribution of the OSE close to but not precisely at the nuclear periphery in WT HCT-116 cells with a much weaker correlation in both the D3 and E4 mutant cells lacking the functional CTCFBS. Since the OSE distribution correlated with the level of *CCAT1* transcription in WT cells, we propose that *CCAT1* transcription provides a net directionality of OSE movements towards a position that is closer to the nuclear periphery. However, as some of the mutant cells showed randomly high *CCAT1* RNA levels without displaying a similarly strong association with the peripheral positioning of the OSE, we also propose that the timing of *CCAT1* transcription should be factored in to explain a net directional mobility of the OSE. The effect of the CTCFBS on *CCAT1* transcription was independently validated by qRT-PCR of both total processed RNA (Fig. 6a, e) as well as newly synthesized RNA (Fig. 6a, f). Since BC21 inhibited *CCAT1* transcription in WT cells, but not in mutant cells (Fig. 6f), we conclude that WNT-activation of *CCAT1* transcription requires a functional *CCAT1*-specific CTCFBS to indirectly facilitate communications between the OSE and *MYC* at the nuclear pehriphery.

## Discussion

This report provides genetic evidence for the multistep principle of gene trafficking to nuclear pores that underlies gene-gating in human cells. Moreover, it identifies a single CTCFBS within the OSE as a key target for WNT signaling to increase the nuclear export rate of *MYC* mRNAs, uncovering a WNT/ß-catenin-AHCTF1-CTCF-eRNA circuit. Although our data overall agrees with other observations linking CTCF to *MYC* regulation and associated pathological growth[12,14], they differ from those data in several fundamental aspects. First, we provide evidence for the role of CTCF in the gating function by mutating its binding site within the OSE, thereby avoiding multiple indirect effects generally associated with the systemic loss of CTCF functions in knock-down experiments. Second, while the *CCAT1*-specific CTCFBS is not essential for OSE-*MYC* interactions to take place in the nuclear interior, it likely facilitates their contact by promoting their proximity to the nuclear periphery/pore. Third, this feature is intimately linked with the CTCFBS-dependent recruitment of AHCTF1, which is a key mediator of the increased *MYC* mRNA export rate[7]. Finally, we also show that the mechanism of repositioning of the OSE to positions close to but not precisely at the nuclear periphery is linked to WNT-dependent *CCAT1* eRNA transcription, a process mediated by CTCF. We argue therefore that the *CCAT1*-specific CTCFBS has a noncanonical function in that it controls mRNA export—a feature not previously documented at any other CTCF binding

site. It is also noteworthy that, in contrast to conclusions discussed in other reports[14,15], reduced *CCAT1* expression/transcription did not affect the overall ability of the OSE to physically communicate with *MYC*. Of note, it has been argued that engineered *CCAT1* eRNA products keep together large CTCF complexes to enable the OSE to communicate with *MYC*[14]. While we cannot yet rule out that such complexes exist to play a role in the gene trafficking process regulated by endogenous *CCAT1* expression, we have been unable to document any effect of RNase treatment on the ability of the *CCAT1*-specific region to bind CTCF. Altogether, there is little or no evidence for a direct function for the *CCAT1* eRNA in anchoring *MYC* to the nuclear pores.

We envisage that the dynamic movement of the OSE from an intra-nucleoplasmic position is initially random, but that the WNT-dependent activation of *CCAT1* eRNA transcription, mediated by the CTCFBS, enables the OSE to reach a position more proximal to the nuclear pore (schematically visualized in Fig. 7). WNT subsequently promotes the recruitment of AHCTF1 to the OSE, which we hypothesize occurs at positions closer to the periphery where CTCF and AHCTF1 are most proximal to each other. Intriguingly, the recruitment of AHCTF1 likely also involves ß-catenin that interacts with TCF4-binding sites flanking the *CCAT1*-specific CTCFBS in a manner unaffected by CTCFBS mutations. Thus, inhibiting the TCF4-ß-catenin interaction by BC21[7] reduced both CTCF-AHCTF1 interactions and their proximities to each other close to the nuclear periphery, and evicted AHCTF1 from chromatin[7]. Taken together, the data are consistent with the interpretation that CTCF collaborates with the ß-catenin-TCF4 complex to stabilize the presence of AHCTF1 on the OSE and that this, in turn, facilitates the ability of the OSE to complete the last stretch and anchor the active *MYC* to the nuclear pores[7].

In summary, we have here provided genetic evidence documenting the gene trafficking/gating phenomenon in human cells. This feature involves a function of a single CTCFBS positioned within the OSE to enable both its ability to migrate to the periphery and anchor *MYC* to the nuclear pore. Although we have not yet examined the potential trafficking of *FAM49B* in similar detail, it appears likely that the CTCFBS is able to either directly or indirectly regulate *FAM49B* expression in a manner similar to *MYC*. This data documents the long-range influence of the *CCAT1*-specific CTCFBS on the post-transcriptional regulation of their expression levels and, by inference, an exacerbation on the overall *MYC* function[25,26]. This model is reinforced by the ability of the CTCFBS within the OSE to confer a WNT-dependent excessive growth advantage to the colon cancer cells. Since the gating of *MYC* is absent in normal human primary colon epithelial cells[7], a path emerges to identify not only diagnostic but also therapeutic strategies that target the effects of

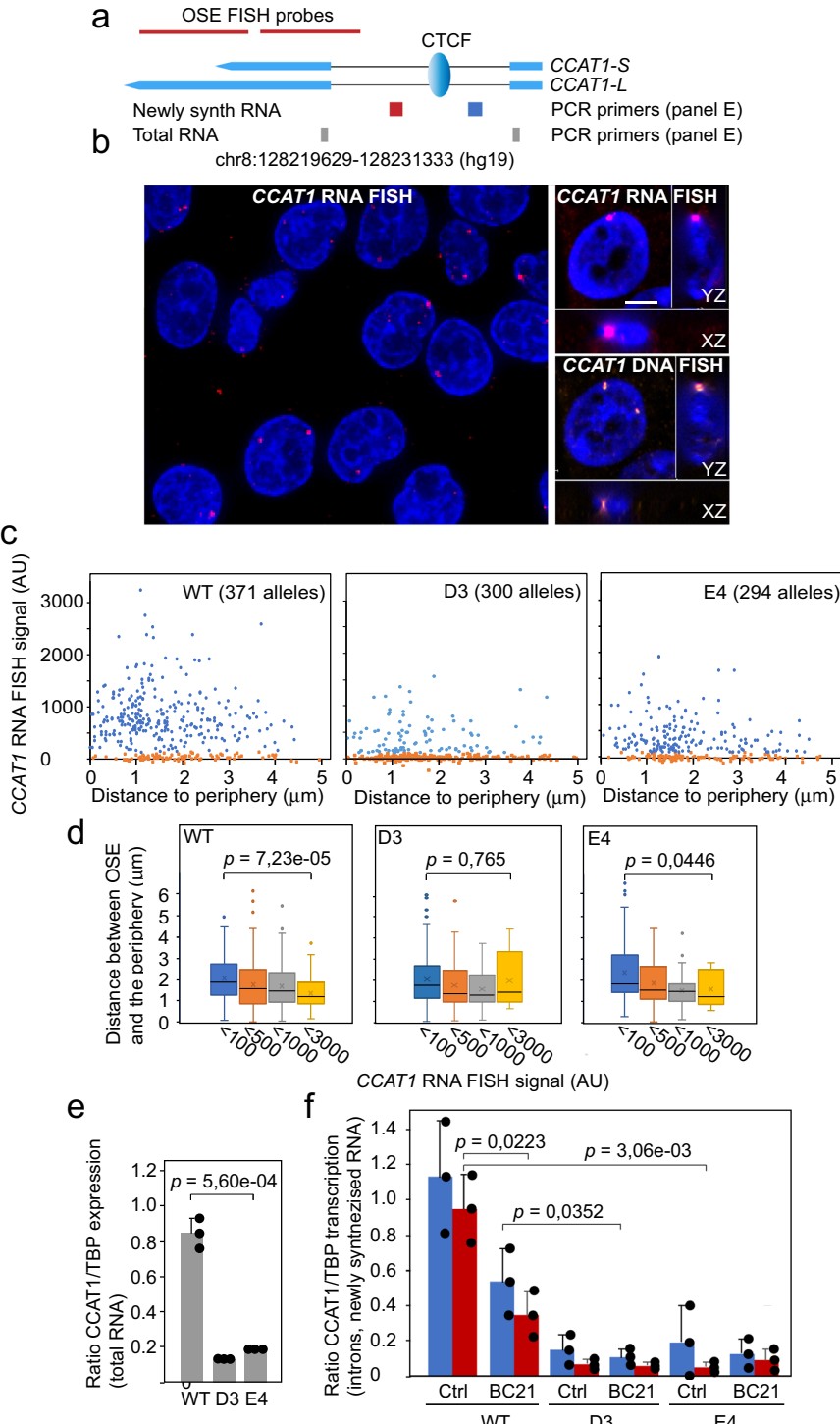

**Fig. 6 WNT activates *CCAT1* eRNA expression via the CTCFBS to promote its juxtaposition to the nuclear periphery. a** Schematic map of the *CCAT1* gene and the position of the CTCFBS and the primers used to assess *CCAT1* expression. **b** Sequential RNA/DNA FISH analyses to score for *CCAT1* expression in WT HCT-116 cells. The left image shows a larger view with the right image representing a focal plane of *CCAT1* RNA and DNA FISH signals. Bar = 3 μm. **c** Quantitation of the RNA FISH signals in relation to the nuclear periphery in WT HCT-116 (371 alleles), D3 (300 alleles) and E4 (294 alleles) cells. Data points at or near background (<100 AU) are marked in orange. **d** Box-and-whisker plots show median values, interquartile ranges and Tukey whiskers (*p* values: two-sided KS test) of the distribution of the OSE in relation to the nuclear periphery, stratified according to the strength of the RNA FISH signal. In instances where no RNA FISH signal could be detected above background, the peripheral distribution was determined by the OSE DNA FISH signal. **e** qRT-PCR analyses of processed *CCAT1* eRNA of total RNA using primers marked as gray in (**a**). **f** qRT-PCR analyses of *CCAT1* transcription using newly synthesized (30 min ethynyl-uridine pulse) RNA as templates and primers positioned upstream (blue) or downstream (red) of the CTCFBS, as marked in panel a). The *p* values indicated in panel d were determined by using the two-sided KS test, whereas the *p* values shown in (**e**, **f**) were determined using the two-tailed Student's *t* test.

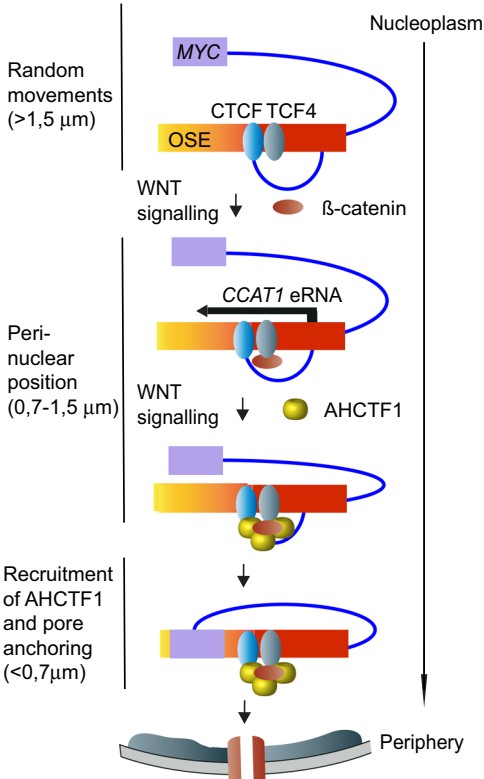

**Fig. 7 A model outlining the role of the CTCFBS to effectuate the WNT-controlled stepwise trafficking of MYC to the nuclear pore.** The migration of the OSE to the nuclear periphery/pore is postulated to occur in three phases: one initially random movement of the OSE in the nuclear interior is followed by a preferential localization of the OSE to a position close to the periphery. This process correlates with WNT-induced activation of *CCAT1* transcription, mediated by the CTCFBS potentially in collaboration with the flanking TCF4 motifs. A third key event is the recruitment of AHCTF1 to the CTCF-complexed OSE, which likely occurs close to the periphery in positions illustrated by proximities between CTCF and AHCTF1. Importantly, this last step is critical for the ability of the OSE to reach the nuclear pore. Prior to the anchoring of *MYC* to the nuclear pore, the potential for direct interaction between the OSE and *MYC* is highest within 0.5 μm from the nuclear pore, paralleled by declining *CCAT1* transcription. This entire process is strongly reduced or absent in mutant D3/E4 HCT-116 cells or normal human primary colon epithelial cells in which the region corresponding to the OSE is unable to efficiently bind CTCF.

pathological *MYC* expression without affecting its normal function. For example, the presence of CTCF within the *CCAT1* gene might provide a diagnostic tool to determine if pathological *MYC* expression involves gene trafficking to nuclear pores. Our results have, moreover, identified several key steps that represent plausible therapeutic targets to downregulate *MYC* expression in cancer. For example, since the binding of CTCF to the OSE represents a focal point in the WNT pathway, inhibitors targeting the partners of CTCF should be considered. Apart from AHCTF1, PARP1 stands out due to its prominent and Olaparib-dependent interaction with CTCF at the nuclear periphery[11].

## Methods

**Cell culture and transfections.** Human colon cancer cells (HCT-116), and primary cultures of normal human colon epithelial cells (HCEC; HCoEpiC, ScienCell, 2950) were maintained and as described before[7]. HCT-116 cells were treated with 10 μM β-Catenin/TCF Inhibitor V, (BC21; Merckmillipore, 219334), or an equivalent amount of the solvent DMSO, for 16 h. Transfection of HCT-116 cells was performed as described[11]. In brief, HCT-116 cells were transfected with 20 nM of CTCF short interfering RNA (siRNA) (Santa Cruz, sc-35124), AHCTF1 siRNA

(Santa Cruz,) or GFP siRNA (Santa Cruz, sc-45924), by using Lipofectamine RNAiMAX transfection reagent (Life Technologies, 13778075) according to the manufacturer's instructions. Following 72 h of incubation, the cells were harvested and the siRNA-mediated downregulation of CTCF was validated by qRT-PCR analysis.

**Editing of the *CCAT1*-specific CTCFBS by CRISPR.** Key sequences within the main CTCFBS within the OSE (chr8:128,219,114-128,219,767) were replaced using the CRISPR/Cas9 technology custom service of Synthego (CA, USA), as illustrated in Fig. 1a. Briefly, specific guide RNA (sgRNA), targeting the CTCFBS within the OSE (see Supplementary Table I), was complexed together with the spCas9 to form a ribonucleoprotein complex (RNP). RNPs and donor DNA were then delivered to the cells via electroporation. The sequence within the CTCFBS was modified from CTCACCATTGGAGGGCATTG to TTCATTATTTTATTTCATTG. Donor DNA sequence: see Supplementary Table I. Following recovery for 2 days, the edits created were evaluated by PCR amplification of the edited site followed by Sanger sequencing. The edited cell pool was used to seed single cells for clonal expansion. Each well seeded is imaged every 2–3 days and rigorously tracked to ensure the population were truly clonal and only the progeny of a single cell. Resulting clones were verified using Sanger sequencing. Two clones (D3 and E4) were selected and expanded. No selection agents were used to enrich for edited populations.

**Off-target whole genome sequencing.** Genome wide off-target sequencing was analyzed according to a modified method of GOTI[22]. Briefly, low-quality reads and adapter were trimmed by Trimmomatic[32]. BWA[33] was subsequently used to align clean reads to the genome and Picard (https://broadinstitute.github.io/picard/) was used to mark duplicates. To reduce the false positive rate, we applied 3 methods to detect the SNV and Indel between D3/E4 and WT, mutect2[34], Strelka2[35] and Lofreq[36]. The overlapping SNVs and Indels were treated as true variants that were further annotated by Annovar[37]. The adjacent 22-bp sequences of the off-target variants were retrieved from the mapping files and blasted with the 22 bp sgRNA (19 bp sgRNA target sequence and 3 bp PAM). High sequence similarities indicated that the off-target variants were sgRNA-associated, while low sequence similarity meant sgRNA-independence[22]. To further screen for potential off-target sites that overlapped with the identified variant we used Cas-OFFinder (http://www.rgenome.net/cas-offinder/).

**ChIP-qPCR.** The cells were fixed with 1% freshly prepared formaldehyde solution as described previously[38]. DNA-protein complexes were immune-purified with antibodies against CTCF (Cell Signaling Technology; CS 2899 S; 20 μl per 10 μg of chromatin DNA), TCF4 (rabbit monoclonal, Cell Signaling Technology, 2569S; 10 μl per 10 μg of chromatin DNA), ß-catenin (Cell Signaling, 8480S; 20 μl per 10 μg of chromatin DNA) or AHCTF1 (Novusbio, NBP1-87952; 2 μg per 6.25 μg of chromatin DNA) using Dynabeads protein G (Thermo Sciences, 10004D)[21]. Following purification of the ChIP DNA (ChIP DNA Clean and Concentrator; Zymo Research, D5205), the associations between the target loci and CTCF and AHCTF1 were quantified by standard qPCR analysis using primer sequences and PCR conditions, as previously described[7] (Supplementary Table I).

**ChIP-seq analyses.** For CTCF ChIP-seq analyses, we used the ChIP-qPCR protocol scaled-up to threefold. The input material used for each of three independent ChIP experiments was ca 19 μg of chromatin DNA. Libraries of the purified immunoprecipitated DNA and un-immunoprecipitated control were generated following the manufacture's recommendations of NEBNext® Ultra™ IIDNA Library Prep Kit (New England BioLabs, US) with index codes added to each sample. Briefly, the genomic DNA was randomly fragmented to an average size of 350 bp. DNA fragments were then end polished, A-tailed, ligated with adapters, size selected and further PCR enriched. The PCR products were purified (AMPure XP system), followed by size distribution by Agilent 2100 Bioanalyzer (Agilent Technologies, CA, USA), and quantification using real-time PCR. The libraries were finally sequenced on NovaSeq 6000 S4 flow cell with the PE150 strategy. The analyses of the ChIP-seq data was performed by nf-core/chipseq pipeline (version 1.2.2)[39]. Briefly, (i) Trim Galore (https://www.bioinformatics.babraham.ac.uk/projects/trim_galore/) was used to trim adapter. BWA was used to perform read alignment[33], (ii) Picard (http://broadinstitute.github.io/picard) was used to mark duplicates, (iii) BEDTools[40] and bedGraphToBigWig[41] were used to create normalized bigWig files, (iv) Phantompeakqualtools[42] was used to calculate strand cross-correlation peak and ChIP-seq quality measures, (v) MACS2[43] was used to call peaks, and (vi) HOMER[44] was used to perform peak annotation. All the read coverages were visualized by the IGV genome browser[45].

**Pulse labeling of RNA.** Newly synthesized RNA was labeled by incubating cells with 0.5 mM (final concentration) 5'-ethynyl uridine (EU, Thermo Scientific, E10345) for 30 min. For analyses of transcriptional activity, the labeled cells were harvested immediately, while in the pulse chase experiment, cells were subsequently washed with 5xPBS followed by incubation with a pre-warmed normal growth medium for 1 h. EU-labeled RNAs were captured using Click-iT Nascent RNA capture kit (Thermo Fisher, C10365) following the manufacturer's

instruction before their conversion into cDNA using SuperScript VILO cDNA Synthesis Kit (Life Technology, 11754050).

**The nuclear RNA export assay**. To determine the ratio between exported cytoplasmic and nuclear RNA, newly synthesized RNAs were pulse-labeled with EU (Thermo Scientific, E10345) as above, followed by the separation of nuclear and cytoplasmic fractions 1 h after chase using the Ambion® PARIS™ system (Thermo Fisher, AM1921) according to the manufacturer's protocol[7]. EU-labeled nuclear and cytoplasmic RNAs were purified and processed to cDNA, as above, for qRT-PCR analyses.

**qRT-PCR analysis of transcription**. The quality of purified RNA samples was assessed (Bioanalyzer, Agilent) before cDNA synthesis (SuperScript VILO cDNA Synthesis Kit, Life Technology, 11754050). All the qPCRs were performed using iTaq Universal SYBR Green Supermix (Bio-Rad, 1725125) on RotorGene 6000 (Corbett Research). The linear range of amplification was confirmed by serial dilution of sonicated genomic DNA from HCT-116 cells. The primer sequences and PCR conditions have been described previously[7] (see also Supplementary Table I).

For the precise measurement of *MYC* and *FAM49B* mRNA levels, one million cells from WT HCT-116 cells, and mutant D3 and E4 clones were counted and washed with warm PBS. Before lysis, 1 μl of diluted (1/10) ERCC ExFold RNA Spike-In Mix (Thermo Fisher Scientific, catalogue number: 4456740) was added to each sample. Cell lysis and RNA purification was performed using RNeasy Mini kit (Qiagen, catalogue number: 74104) following the manufacturer's protocol. RNA was eluted in 50 μl of nuclease free water, and RT-qPCR was performed using a two-step protocol as follows: 3 μl of each sample was reverse-transcribed using the SuperScript™ VILO™ cDNA Synthesis Kit. cDNA samples were diluted (1/50) in nuclease free water and qPCR was performed in 20 μl reaction containing 10 μl of iTaq Universal SYBR Green Supermix (Biorad, catalogue number: 1725124), 6 μl nuclease free water, 1 μl of both 10 mM forward and reverse primers (Supplementary Table I), and 2 μl of diluted cDNA. The qPCR reaction was performed in Corbett Rotor-Gene 6000 HRM Real Time PCR Machine.

**Computer modeling of *MYC* mRNA cytoplasmic levels over time**. To model the gradual increase of cytoplasmic levels of *MYC* mRNA over time from an initial time point, defined as α for the earliest measurable transcription rate, we followed a mathematical model which outlines the dynamics of nuclear ($X$) and cytoplasmic ($Y$) mRNAs[46], as we have described earlier[7].

**RNA-seq**. RNA-seq library constructions were generated with standard Illumina TruSeq Stranded mRNA kit with Poly-A selection and samples were sequenced on NovaSeq6000 (NovaSeq Control Software 1.7.0/RTA v3.4.4) with a 51nt(Read1)-10nt(Index1)-10nt(Index2)-51nt(Read2) setup using "NovaSeqStandard" workflow in "SP" mode flowcell. Data analysis was performed by nf-core RNA-seq pipeline (v.3.0)[39] with default parameter. In brief, adapters and low-quality reads were filtered by trimgalore (0.6.6)(https://github.com/FelixKrueger/TrimGalore). Clean reads were aligned to the GRCh37 human reference genome[47] and ERCC RNA spike-in fasta (https://www.thermofisher.com/se/en/home.html) with hisat2(2.2.0)[48]. Genome-wide coverage output in BEDGRAPH format were generated by Bedtools genomecov subcommand (V 2.29.2)[40]. UCSC bedSort and bedGraphToBigWig (V377) were used to Create bigWig coverage files[41]. Integrative Genomics Viewer[45] was used to view the read coverage.

**DNA FISH analyses**. The DNA FISH probes were prepared from a pool of PCR products spanning 8 to 10 kb regions of Hind III sites encompassing the *MYC* promoter and gene body (chr8:128,746,000-128,756,177) and the OSE (positioned at chr8:128,216,526-128,225,855), respectively (Supplementary Table I). The probes were labeled with either biotin-16-dUTP (Roche, 11093070910) or digoxigenin-11-dUTP (Roche, 11573179910), using a Bioprime Array CGH kit (Life Technologies, 18095011), and hybridized to formaldehyde cross-linked cells as described before[7]. To ensure that the PCR probes identified the correct genomic region surrounding the OSE we used the BAC clone CTD-3066D1. Following hybridization, the cells were incubated with primary anti-biotin antibodies (Cell Signaling, 5597S; diluted 1:200) and anti-digoxigenin antibodies (Roche, 11333062910; diluted 1:200) overnight at 4 °C. Subsequently, the cells were washed with 0.05% Tween 20 in 1× PBS, incubated with secondary antibodies for 1 h at room temperature and then washed with 0.05% Tween 20 in 1× PBS again. Finally, the cells were counterstained with DAPI solution (ThermoFisher Scientific, 62248) and mounted in Vectashield mounting medium (Vector Labs, H-1900). To ensure the identity of hybridization signals resulting from the PCR-amplified probes, BACs for the OSE and MYC regions were routinely included, as has been described before[7].

**RNA FISH analyses**. The *CCAT1* RNA FISH probes were generated from two PCR products spanning a region within the OSE (positioned at chr8:128,216,526-128,225,855, Supplementary Table I) and labeled with biotin-16-dUTP (Roche, 11093070910), as described above. RNA FISH analyses were performed as previously described[7].

**In situ proximity ligation assays (ISPLA)**. The ISPLA analyses were performed using cells fixed with 1% paraformaldehyde in Phosphate Buffered Saline following a previously described protocol[11]. For the CTCF-AHCTF1 ISPLA, purified mouse anti-CTCF (BD Biosciences, 612148; diluted 1:40) and rabbit polyclonal anti-AHCTF1 (Novus Biologicals, NBP1-87952; diluted 1:100) were used as primary antibodies. For CTCF-NUP133 ISPLA, purified mouse anti-CTCF (BD Biosciences, 612148; diluted 1:40) and NUP133 (Abcam, ab155990; diluted 1:100) were used. Oligonucleotide-conjugated anti-rabbit and anti-mouse antibodies were then used to generate rolling circle amplification, as described previously[11]. Samples that lacked the primary antibodies served as background control. Quantitation of the ISPLA signals (marked with the Cy3 fluorophore) inside the nuclei (counterstained with DAPI) was performed on software Imaris v.8.1.2 (Bitplane, Switzerland).

**Wide-field microscopy**. Cell imaging and the generation of optical sections in 3D were carried out on a Leica DMi8 microscope, equipped with a HC PL APO 63X NA 1.4 oil objective and DFC9000 camera, using the Instant Computational Clearing Method of the Thunder Imaging System (Leica Microsystems). Stacks were taken using the software system optimized intervals in the z axis. Pictures were analyzed with the use of the Imaris or Leica Application Suite X (LasX) softwares. Due to the limitations in the resolution of the fluorophores (with CY3 at 239,6 nanometer), the distance data were stratified using 240 nanometers as the first cut-off.

**Co-immunoprecipitation and western analysis**. All co-immunoprecipitation experiments were performed using the Nuclear Complex Co-IP kit, following the protocol provided by the manufacturer (Active Motif, 54001). Briefly, 250 ug of protein from nuclear lysates were incubated with Dynabeads protein G (Thermo Fisher Scientific, 10004D) for 1 h at 4 °C. Following pre-clearing, the samples were incubated O/N at 4 °C with the anti-CTCF (Abcam, ab37477, mouse; 2.5 μg per 250 μg of input protein) or normal mouse IgG (Santa Cruz, sc-2025; 2.5 μg per 250 μg of input protein). The immunoprecipitated samples were analyzed with a Simple Western assay using the WES™ system (ProteinSimple, Bio-Techne). The following antibodies were used for the Western analysis: anti-CTCF (Cell Signaling, 2899S, rabbit; diluted 1:50), anti-NUP133 (Abcam, ab155990, rabbit; diluted 1:25), anti-AHCTF1 (Novus Bio, NB600-238, rabbit; diluted 1:50), anti-β-catenin (Cell Signaling, 8480S, rabbit; diluted 1:100) and anti-TATA binding protein (Abcam, ab51841, mouse; diluted 1:50). The relative amount of each protein was quantified via the peak areas detected in the chemiluminescence electropherogram generated by the Compass for SW software (ProteinSimple, San Jose, CA), following the default settings. A standard curve based on the serial dilutions of the input was used to estimate the absolute amount of protein in each sample. Finally, the recovery of input for each identified interacting protein was calculated through normalization to the percentage recovery of CTCF. All antibodies were approved of only if they detected the correct bands upon WES/JESS analyses.

**Nodewalk**. The analyses of physical interactions between *MYC* and the OSE were performed using an anchor representing the *MYC* gene, as has been previously described[7,24].

**Statistics**. Most of the statistical analyses were performed by using the two-tailed Student's *t* test of the average of three independent experiments in each of these experiments. Error bars indicate mean ± standard deviation. To assess the significance of the distribution of OSE alleles in relation to the periphery, the two-sample, two-tailed Kolmogorov–Smirnov (KS) test was applied at a significance level of 0.05. Since the KS test is a nonparametric test used to compare two unpaired sets of data, it detects the difference between the two distributions.

**Reporting summary**. Further information on research design is available in the Nature Research Reporting Summary linked to this article.

## Data availability

The data that support this study are available from the corresponding author upon reasonable request. The genomic data reported in this paper has been deposited to general databases with the following accession numbers: For genomic sequences SRA accession nr: PRJNA756713, for ChIP-seq GEO accession nr: GSE184106 and for RNA-seq data GEO accession nr: GSE184103. Source data are provided with this paper.

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

## Acknowledgements

This work was supported by the Swedish Research Council (VR 2016-03108), the Swedish Childhood Cancer Fund (PR2017-0132), the Swedish Cancer Society (CAN 2016/708), the Lundberg Foundation (2018-0138), Karolinska Institutet, the Novo Nordisk Foundation (NNF16OC0021512), The Cancer Society in Stockholm (Cancerföreningen, 2018-2019), the MARIE Skłodowska-CURIE ACTIONS (Chromatin 3D) and the KA Wallenberg Foundation (KAW 2017.0077). The authors acknowledge support from the National Genomics Infrastructure in Stockholm funded by Science for Life Laboratory, the Knut and Alice Wallenberg Foundation and the Swedish Research Council, and SNIC/Uppsala Multidisciplinary Center for Advanced Computational Science for assistance with massively parallel sequencing and access to the UPPMAX computational infrastructure.

## Author contributions

I.C. sequenced the mutant OSEs and contributed to the expression, mRNA export analyses, co-culture experiments, and RNA-seq library constructions; I.T. produced most of the ChIPs and co-immunoprecipitation analyses, HD contributed with expression, and mRNA export analyses; J.P.L. performed most of the RNA/DNA FISH experiments and contributed to co-ip experiments; A.L.R. and B.S. contributed to the DNA FISH analyses, F.B.C. performed the ISPLA experiments; S.W. analyzed ChIP-seq data and screened off-target sites, J.V. performed the Nodewalk experiments, C.D.M.L. contributed to the ChIP analyses; D.B. analysed the Nodewalk and ChIP-seq data and MM contributed to the co-immunoprecipitation experiments and designed the sgRNA, R.M. performed the simulation analysis and contributed to the RNA-seq analyses; and A.G. conceived, supervised, planned the experiments, and wrote the paper.

## Funding

## Competing interests

The authors declare no competing interests.
