## [Peer Review File · Nature Communications]

REVIEWER COMMENTS

Reviewer #1 (Remarks to the Author):

This manuscript is a follow-up study of their previous finding that “WNT signaling and AHCTF1 promote oncogenic MYC expression through super-enhancer-mediated gene gating” (Scholz et al. Nature Genetics), which extended to the investigating the role of CTCF binding in OSE in the same model system. They claimed that CTCF binds to a single CBS, directed in the WNT-dependent trafficking of the entire OSE to the nuclear pore from intra-nucleoplasmic positions. When the OSE reached a perinuclear position, triggered by CTCFBS-mediated CCAT1 eRNA activation, its final stretch (<1µm) to the nuclear pore required the recruitment of AHCTF1, a key nucleoporin, to the CTCFBS. Although it seems to be an interesting study to reveal the MYC regulation under 3D nuclear architecture at first glance, I then have significant incremental concerns about the uncertain quality control and interpretation of the results, which dampened my enthusiasm to recommend for publication at the current status. I hope that the authors will find the enclosed comments helpful.

Major comments

1. The entire work relies on two CBS-mutant clones (D3 and E4) derived from CRISPR-targeted HDR. Based on the recent elegant work from Dr. David Spencer (Leukemia 2021), single-cell-derived clones show significantly high variation regarding transcriptional regulation of target genes. At least two more independent clones should be included to confirm the phenotype. Also, I am not sure how the control was generated, either from the wt pool or it is indeed a pool? The best-matched controls should be derived from the targeted pool and sequenced as the WT allele.
2. Following the first question, what’s the expectation of total transcriptional changes of MYC upon CBS mutation? Given that CTCF protein is responsible for enh/pro looping, dimerization, RNA-interaction, and DNA binding, does mutating the CBS in the OSE reduce MYC expression? If so, how could the single-cell-derived clones derived from the very beginning? If not, what’s the explanation?
3. It is known that the CTCF motif (16-20bp) is relatively conserved among different species. Therefore, targeting the single CBS by CRISPR may cause the off-targeting issue in other genomic CBS sites. Did the author examine a genome-wide scale carefully? Even targeting is specific, the 8 bp switch from C to A may still cause the mutation of other TFs on top of CTCF’s binding site. For instance, YY1 was usually co-binding in many CTCF binding sites recognizing the “ATGG” motif. Here I am observing the same motif in OSE as well. I guess how to relate the function of CBS-mutants to CTCF specifically is the keep question, which was not answered with a clean strategy.
4. What’s the 3D interaction between MYC promoter with OSE and enhancer? It should not be difficult to examine by 3C PCR or Capture-C.
5. The measurement of “recruitment to nuclear pore” is a single assay, which should be complemented with an additional strategy with dynamic imaging or pulldown assay in fig 2.
6. Why were only data shown from clone E4 in Figures 3 and 4? How about D3?
7. In figure 4, the additional information of DNA FISH with MYC promoter and OSE should be included on top of CCAT1.
8. The gating of MYC through nuclear pore probably most important to control the subcellular and translocation of nascent RNAs in cytosol or nucleus. The authors should quantify this process between WT and CBS-mutant cells.

Minor comments:

1. In detailed information on how WT control was established is missing in the method.
2. BC21 drug effect was not confirmed or shown in the study.
3. GSM749690 is publicly available CTCF-ChIP-seq data of K562 from ENCODE, which is irrelevant to HCT116 from the current study. The author should provide ChIP data between WT and CBS-mutant to rule out the possibility of an off-target effect on a genome-wide scale.
4. Statistical analysis and justification are not provided at all in the method session, which is problematic for me to judge the data quality.

Reviewer #2 (Remarks to the Author):

Chachoua et al. aim to dissect the mechanism of « gene gating » as a process of MYC overexpression in colon cancer cells. "Gene gating" facilitates gene overexpression via their juxtaposition to the nuclear pore in order to rapidly export mRNAs in the cytoplasm. They showed that a CTCF binding site (CTCFBS) within an oncogenic super-enhancer (OSE) of MYC is involved in the recruitment of the nuclear-pore factor AHCTF1 in a WNT signaling pathway dependent manner in order to promote pathological nuclear export of MYC mRNAs. This study provides important insights into the gene gating process in human cells, and in pathological conditions. This could open new avenues in defining new therapeutic strategies. However, some questions should be addressed.

1) HCT-116 and HCEC cells could be briefly introduced in the first section of the results.

2) Figure 1:

- Fig1A: the scheme is not clear. What are the orange and red boxes? Although the numbers below the scheme must be genomic coordinates, it is not clear what they represent. OSE and CCAT1 may be shown.

- Fig1B: The authors use 2 mutant clones for OSE (D3 and E4). Are the 2 OSE alleles mutated? Do these 2 clones harbor the exact same mutations? Did the authors check for potential off-targets?

- Fig1G: What are the mutant CTCFBS? D3 or E4 or else?

- The statistics are not clear. In the different panels, it is not clear whether they are calculated only of E4 clone or for both mutant clones combined.

- It is not clear why there are 3 copies of MYC/FAM49B alleles in WT cells. What are exactly these WT cells? Do they thus represent the ideal control?

3) Figure 2:

- This is a detail but, although it seems clear that these experiments are performed in HCT-116 cancer cells, it may be mentioned.

- Fig2A-B:

i) The ISPLA assay shows proximity between CTCF and AHCTF1 in absence or presence of BC21 inhibitor. A control in WT cells could have been added there.

ii) The ISPLA signals should be localized close to the nuclear periphery. Their distance to the nuclear periphery could have been measured and shown like in Fig2H.

iii) One question is the localization of the OSE in the nuclear space in WT and cancer cells? Are they, by default (in WT cells), localized in particular regions of the nucleus?

iv) Is there a difference in the amount of AHCTF1 proteins in cancer versus WT cells? If there is more AHCTF1 proteins in cancer cells, its association with CTCF may be aspecific and due to its overrepresentation. This control may be added in the study. In a similar way, is there a difference in the average number of nuclear pores in cancer versus WT cells?

- Fig2C/D/F: co-immunoprecipitation CTCF-AHCTF1 is shown in cells with WT OSE, in absence or presence of BC21. A control in mutant-OSE cells may have been added, as for the ChIP experiment (Fig2E).

- Fig2J: The authors describe the model they elaborated with the different experiments.

i) They wrote that CTCFBS recruits AHCTF1. Is it really CTCFBS or CTCF bound to CTCFBS? The role of CTCF is not described in the model.

ii) They argue that NUP133 and b-cat are indirectly recruited to the OSE. I may have missed a point but I am not sure that this indirect recruitment is formally shown.

iii) The authors showed, with the BC21 inhibitor, that the WNT signaling pathway is involved in the recruitment of AHCTF1 to the OSE. Is the WNT pathway involved in the recruitment of CTCF on the CTCFBS? In other words, did the authors perform a CTCF-ChIP in presence of BC21?

iv) As the WNT pathway is involved in the recruitment of AHCTF1, I am confused by the statement that CTCFBS indirectly recruits b-cat, which is part of the WNT pathway.

4) Figure 3:

The authors explored the nuclear localization of the MYC alleles in HCT-116 cells with WT versus mutated CTCFBS. They found that the proximity of both OSE and MYC to the nuclear periphery is reduced in mutated-CTCFBS cells.

- Are these differences in proximity to the nuclear periphery statistically significant?

- It could have been interesting to analyze the distance to the nuclear periphery of OSE and MYC in WT colon cells, as well as in HCT-116 cells treated with BC21 and/or siAHCTF1 in order to control the role of the WNT-pathway and AHCTF1 in the localization of MYC and OSE.

5) Figure 3/4:

The authors further showed that the mutations induced in CTCFBS, although important for the recruitment to the nuclear pore, is not involved in OSE-MYC interactions. Moreover, expression of the CCAT1 eRNA close to the periphery does not seem to be involved in the OSE-MYC interactions. Do the authors have any hypothesis on the mechanism of OSE-MYC interactions?

6) Discussion:

- The authors hypothesize that the recruitment of AHCTF1 to the OSE occurs at perinuclear positions. Would it be possible to perform combined immunofluorescence of AHCTF1 with DNA FISH for OSE in order to validate this hypothesis?

- Do the authors have any hypothesis on which of the WNT signaling pathway component could be directly involved in this process?

Reviewer #3 (Remarks to the Author):

Summary

Chachoua et al. describe a mechanism where a single CTCF binding site within an oncogenic super enhancer (OSE) facilitates the trafficking of the MYC gene to the nuclear periphery to increase MYC mRNA export following WNT signaling, specifically in the HCT116 colon cancer cell line, but not in normal colon epithelial cells. To study this mechanism, the authors mutated the CTCF binding site within the OSE and studied how the mutation affects MYC's gene expression, mRNA export, ability to recruit AHCTF1 (nucleoporin that's essential for gene gating), and association with the nuclear periphery. The CTCFBS is in an eRNA gene, CCAT1, which mechanistically they show is important to tether MYC to the nuclear pore for efficient gene gating.

Major Concerns

1) All of the experiments are carried out in a single colon cancer cell line. Demonstrating that this is relevant to colon cancer and/or other cancer cell lines is important to assess the significance of the report.

2) The claim that a single CTCFBS in the OSE is the relevant controller for MYC gene gating should include controls mutating other CTCF binding sites in/relevant to the MYC gene to rule out general disruption of chromatin architecture resulting in unrelated loss of MYC gene gating.

3) For relevance to Wnt signaling, other methods in addition to the BC21 inhibitor should be used.

4) In addition to the co-culture experiment followed by qPCR to show a growth advantage in figure 1, a proliferation assay in the mutant cells would bolster the claim that the CTCF-binding site confers an increase in growth advantage. In addition, an analysis of the mutant having a reduced ability to respond to WNT signaling would strengthen the report.

5) The co-IP experiment in figure 2 shows the interaction between CTCF, AHCTF1, and Nup133, however, to be more confident of the location of this interaction, an In-Situ PLA experiment would be a good addition.

6) For the 3D FISH experiment in figure 2 where AHCTF1 knockdown resulted in the OSE further from the nuclear periphery, statistical analysis of the 3D FISH experiment should be included.

7) Figure 3 needs statistical significance determinations.

8) In figure 3, the E4 mutant showed that MYC's connection to the OSE was unaffected while WT had a significant change in association. Showing that the D3 mutant was unaffected as well would add confidence to the claim.

9) Addition of the D3 mutant to figure 4 experiments would add strength.

10) The strong claim that this study was the first genetic evidence of gating/trafficking phenomenon in human cells is not supported by current literature. The inclusion of Kadota et al., nature

communications (2020) outlines an epigenetic memory at the nuclear pore which includes the gene gating mechanism.

Minor Concerns

- 1) In figure 3B, the y-axis says the measurement is a percentage but has units of " μm ".
- 2) More clear experimental descriptions would improve the read of the paper.

POINT-BY-POINT RESPONSES TO THE REVIEWER'S COMMENTS

On behalf of all the co-authors, I would, first of all, like to thank all Reviewers for several constructive comments that clearly improved our manuscript. We have responded to all the comments to the best of our abilities. The result is that a considerable amount of new data has been added. To better visualize the data, the manuscript has undergone a major re-organization with the transfer of some of the Supplementary Figure panels to the main Figures. To avoid too compact Figure panels, these have been split as follows:

Original manuscript	Revised manuscript
Figure 1	Now Figures 1 and 2
Figure 2	Now Figures 3 and 4
Figure 3	Now Figure 5
Figure 4	Now Figure 6
Figure 5	Now Figure 7

I would also like to emphasize that new data has been added to all Figures with the exception of Fig. 7 (previously Fig 5) and Supplementary Fig. 6

Reviewer #1 (Remarks to the Author):

This manuscript is a follow-up study of their previous finding that “WNT signaling and AHCTF1 promote oncogenic MYC expression through super-enhancer-mediated gene gating” (Scholz et al. Nature Genetics), which Supplementary to the investigating the role of CTCF binding in OSE in the same model system. They claimed that CTCF binds to a single CBS, directed in the WNT-dependent trafficking of the entire OSE to the nuclear pore from intra-nucleoplasmic positions. When the OSE reached a perinuclear position, triggered by CTCFBS-mediated CCAT1 eRNA activation, its final stretch (<1 μ m) to the nuclear pore required the recruitment of AHCTF1, a key nucleoporin, to the CTCFBS. Although it seems to be an interesting study to reveal the MYC regulation under 3D nuclear architecture at first glance, I then have significant incremental concerns about the uncertain quality control and interpretation of the results, which dampened my enthusiasm to recommend for publication at the current status. I hope that the authors will find the enclosed comments helpful.

Major comments

1.The entire work relies on two CBS-mutant clones (D3 and E4) derived from CRISPR-targeted HDR. Based on the recent elegant work from Dr. David Spencer (Leukemia 2021), single-cell-derived clones show significantly high variation regarding transcriptional regulation of target genes. At least two more independent clones should be included to confirm the phenotype. Also, I am not sure how the control was generated, either from the wt pool or it is indeed a pool? The best-matched controls should be derived from the targeted pool and sequenced as the WT allele.

Our response:

The recent paper by the Spencer group is indeed impressive. However, we would like to point out that the reported individual variation in CRISPR-generated clones was observed only for primary AML cultures, which are expected to be very heterogenous. Conversely, when the authors of this report examined an established AML cell line, little or no expression variation could be observed as a result of editing a set of selected CTCF binding sites. Although it is of course likely for cloned cell lines, such as the HCT-116 cell line used in our study, to drift both genetically and epigenetically, the variants providing a growth advantage will likely take over the cell population to continuously reduce heterogeneities under constant culture conditions. We also want to highlight that the Spencer report often used only one clone for subsequent in-depth analyses, whereas we have used two clones throughout the manuscript. Incidentally, no off-target screens were included in this study. Finally, our manuscript and the Spencer publication use the same definition of the WT cell population, i.e. the parental cell population used for the editing experiments.

We do not consider the use of WT cell pool that undergone CRISPR editing but lack the mutant OSE donor sequence a good control due to the heterogenous and undefined/uncontrollable exposure of cells to the CRISPR reagents in a cell population. It is thus very difficult to determine if the lack of proper donor DNA integration is due to either insufficient transfection of the CRISPR reagents into those particular cells or to other underlying mechanisms involving, e.g., defects in DNA repair that might provide selectable features against integration. Instead, to support our conclusions we have focused on i) identifying potential genetic and epigenetic off-target effects in the mutant D3 and E4 clones and ii) identifying the molecular players and underlying mechanisms that explain the phenotype of the mutant clones.

2. Following the first question, what's the expectation of total transcriptional changes of MYC upon CBS mutation? Given that CTCF protein is responsible for enh/pro looping, dimerization, RNA-interaction, and DNA binding, does mutating the CBS in the OSE reduce MYC expression? If so, how could the single-cell-derived clones derived from the very beginning? If not, what's the explanation?

Our response:

We would like to point out that the original manuscript has already incorporated the unexpected finding that the mutation of the CTCF binding site within the OSE does not reduce the level of *MYC* transcription. There was thus no difference in *MYC* transcriptional rates between the WT HCT-116 cells and the D3 and E4 clones, as was shown in the Supplementary Figure 1 of the original manuscript. To better highlight this data, the transcriptional rates of the *MYC* and *FAM49B* genes in the WT HCT-116 and mutant cells have now been moved to the new Figure 2. The manuscript thus identifies a major role for this CTCF binding site within the OSE in increasing the nuclear export rate of *MYC* mRNAs *via* gene gating mechanism, which enables the escape of *MYC* mRNAs from the high degradation rate in the nucleus compared to the cytoplasm. The result of the mutation is thus reduced levels but not absence of total cellular *MYC* mRNA. Although the WT HCT-116 cells have a clear growth advantage over the mutant cells when co-cultured, the mutant cells do grow and thus enable the generation of single clones upon CRISPR editing.

3. It is known that the CTCF motif (16-20bp) is relatively conserved among different species. Therefore, targeting the single CBS by CRISPR may cause the off-targeting issue in other genomic CBS sites. Did the author examine a genome-wide scale carefully? Even targeting is specific, the 8 bp switch from C to A may still cause the mutation of other TFs on top of CTCF's binding site. For instance, YY1 was usually co-binding in many CTCF binding sites recognizing the "ATGG" motif. Here I am observing the same motif in OSE as well. I guess how to relate the function of CBS-mutants to CTCF specifically is the keep question, which was not answered with a clean strategy.

Our response:

To prevent CTCF binding to the OSE, we have here implemented a strategy that is very similar to the efficient strategy we used to mutate CTCF binding sites in a mouse knock-out model^{1,2}. We agree with the reviewer, however, that even point mutations of key bases within a recognition motif of one factor might inadvertently affect the binding of other transcription factors to this region, which could contribute to the phenotype of the mutant clones. To further strengthen our conclusion that it is indeed CTCF that regulates *MYC* gating *via* binding to the CTCF binding site within the OSE, we have performed additional experiments.

1. A key new data is that down-regulation of cellular CTCF levels by siRNA reduces the binding of both CTCF and AHCTF1 to the *CCAT1*-specific CTCF binding site in WT cells, and does so to a similar extent as observed in the D3/E4 cells.

2. Using ChIP-qPCR, we have also examined how the CTCFBS mutation affects the binding of additional key factors, such as the binding of β -catenin-TCF4 complex, to the OSE, which we have previously shown to regulate *MYC* gating (ref nr 7) and have binding sites flanking the edited CTCFBS. The results show no reduction in β -catenin or TCF4 binding to the OSE neighboring the mutated *CCAT1*-specific CTCF binding site in the D3 and E4 clones compared to the WT HCT-116 cells. This result is included in a new Supplementary Fig. 4A,B. These observations have thus improved our understanding of the role of WNT signaling in *MYC* gating, as discussed in a new Figure 4, and highlight that CTCF binding to a specific site within the OSE is necessary for the efficient induction of the gating of *MYC* by WNT.

3. We agree with the reviewer that YY1 is a well-known partner of CTCF⁵, and we do not rule out a role for this factor in the gating process. However, this is beyond the current manuscript that focuses on the requirement of CTCF binding to the OSE in the gene gating process. Furthermore, we would like to point out that only four bases of the 9 bp consensus YY1 binding motif (AAnATGGCG) overlap with the CTCFBS. It is also noteworthy that cis elements with longer motifs (CAAATGGCGGC) display a much stronger affinity to YY1³. We also note that the four bases alluded to by the Reviewer are expected to occur on average every few kb in the whole genome, i.e. almost a million such sites, which far exceeds the number of observed genome wide YY1 binding sites⁴. All of this suggests that any partial sequence similarity to the YY1 motif is unlikely to influence the interpretation of our manuscript.

4. To rule out potential off-target effects that might disrupt other conserved CTCF binding sites in this region that could influence the gating process we have performed CTCF ChIP-seq experiments on the WT HCT-116 cells and the D3 and E4 clones. The results show unchanged CTCF binding frequencies to the OSE-*MYC* region in the WT, D3 and E4 clones except for the edited CTCFBS. This data is included in Fig. 1D of the revised manuscript.

5. Finally, the genomes of the WT and mutant cells have been sequenced and analysed for off-target effects. In the manuscript we write that we “ - adapted a genome wide off-target detection pipeline modified from GOTI²² (Supplementary Fig. 1A). Allowing for a filter of 100% sequence similarity for a blast word size of ≥ 12 bases, we found no sgRNA-associated target for the D3 and E4 cells in the blast results. We further analysed the potential off-targets by using Cas-OFFinder (<http://www.rgenome.net/cas-offinder/>) allowing 5 mismatches in the settings to directly align the sgRNA with the genome. No overlap was found between the variant location and potential off-target sites to indicate that the editing process did not as such generate genome wide variants. However, we did find a total of 91 indels and 10 SNVs that were shared between the D3 and E4 cell clones. Two of the indels that were common to both cell clones mapped to the vicinity of CTCFBSs on chromosome 8 and 22. ChIP-seq analyses showed that these changes did not antagonize CTCF binding (Supplementary Fig. 1B). Since neither of these regions interacted with *MYC* or the OSE (not shown) using the Nodewalk assay, it is unlikely that they are involved in the *MYC* gating process.”

4. What's the 3D interaction between *MYC* promoter with OSE and enhancer? It should not be difficult to examine by 3C PCR or Capture-C.

Our response:

The interaction frequency between the OSE and the *MYC* gene was already included as a bar diagram in Figure 3 in the original version of our manuscript. Just in case, we have in the revised manuscript included both bar diagrams and graphs of physical interactions between the *MYC* anchor and the oncogenic super-enhancer in a new Figure 5E,F. This data was generated from three independent experiments by the ultra-sensitive Nodewalk technique, which is a 3C-based method described in our recent Nature genetics and Nucleic Acid Research papers referred to in both the original and revised manuscripts (ref nrs 7 and 24 in the revised manuscript). We have thus found no reduction in interaction frequency between the mutant OSE and *MYC* in the D3 and E4 clones compared to the WT HCT-116 cells, demonstrating that the *CCAT1*-specific CTCF binding site is not directly responsible for the OSE-*MYC* interaction.

5. The measurement of “recruitment to nuclear pore” is a single assay, which should be complemented with an additional strategy with dynamic imaging or pulldown assay in fig 2.

Our response:

Such data, resulting from additional strategies, has already been published in our recent nature genetics paper (ref nr 7 in the revised manuscript). In that report, we used both ChIP pulldowns to show specific binding of NUP133, but not NUP153, to the OSE and the ChrISP (chromatin *in situ* proximity) technique to visualize that the oncogenic super-

enhancer and NUP133 are very close to each other when proximal to the nuclear pore and that this feature is antagonized when AHCTF1 expression is reduced. In Figure 2 of this nature genetics paper (ref nr 7) we were also able to directly visualize the anchoring of the oncogenic super-enhancer to a nuclear pore with a resolution of $<160\text{\AA}$. In the current manuscript, we have relied on the DNA FISH assay because of the clear, significant difference we observed in the sub-nuclear localization of the OSE and *MYC* regions in the presence of mutated *CCAT1*-specific CTCF binding site compared to the WT OSE. As the gating principle requires proximity to nuclear pores/periphery, we have used the versatile DNA/RNA FISH assay to examine if the mechanism of action of the *CCAT1*-specific CTCF binding site involves the regulation of the trafficking of the OSE to sub-nuclear positions close to the nuclear periphery. Moreover, we have performed ChIP assays showing the strongly reduced binding of the nuclear pore component AHCTF1 to the mutant CTCF binding site.

6. Why were only data shown from clone E4 in Figures 3 and 4? How about D3?

Our response:

The reason for the exclusion of the D3 clone data was that it seemed repetitious to include very similar data in all aspects. All the in-depth data of the edited clones including the D3 cells, is, moreover, available in the Source data file. In the revised manuscript we have included processed data (including statistical analyses) that compares the WT cells with both the D3 and E4 clones in more detail. The statistically significant data presented in the new Figures 5 and 6 (previous Figures 3 and 4) show that both clones behave very similarly in comparison to WT cells.

7. In figure 4, the additional information of DNA FISH with MYC promoter and OSE should be included on top of *CCAT1*.

Our response:

The *CCAT1* image in Figure 4 represents a magnified portion of the oncogenic super-enhancer. While the coordinates of the DNA FISH probes are included in the legend of the new Figure 5, as they were in the original manuscript, the positions of the OSE and *MYC* DNA FISH probes have now been schematically illustrated in a new Fig. 5A.

8. The gating of *MYC* through nuclear pore probably most important to control the subcellular and translocation of nascent RNAs in cytosol or nucleus. The authors should quantify this process between WT and CBS-mutant cells.

Our response:

This very important data was presented in the original manuscript in Fig. 1D (Fig. 2A in the revised manuscript) and is indeed the key to the interpretation that the *CCAT1*-specific CTCFBS controls the nuclear export of *MYC* mRNAs. Moreover, we have also analysed the effect of the β -catenin antagonist, BC21, on the nuclear export rate of *MYC* mRNAs in WT HCT-116 cells and the D3 and E4 clones. Fig 1D in the original manuscript (Fig. 2A in the revised manuscript) shows that the nuclear export rate is reduced by BC21 in WT HCT-116, whereas no further reduction in the export rate was observed in the D3 and E4 clones

where the nuclear export rate of *MYC* mRNAs is already low. These results highlight that the WNT-dependent activation of the *MYC* gating process requires the *CCAT1*-specific CTCFBS.

Minor comments:

1. In detailed information on how WT control was established is missing in the method.

Our response:

The definition of what we mean with WT HCT-116 cells, i.e. the parental cell line, has been clarified in the text of the revised manuscript.

2. BC21 drug effect was not confirmed or shown in the study.

Our response:

The effect of this drug, which antagonizes the formation of β -catenin-TCF4 complexes, was documented in our recent nature genetics report (ref nr 7 in the revised manuscript) and was referred to in the original version of our manuscript. Briefly, BC21 both evicts β -catenin (and NUP133 as well as AHCTF1) from the region immediately downstream of the *CCAT1*-specific CTCFBS that carries a TCF4 binding site (Supplementary Fig. 4A,B) and reduces the proximity between TCF4 and β -catenin. We are routinely ensuring that the BC21 effect is reproducible when performing new experiments by examining its potential to evict β -catenin from the OSE (ref. nr 7 in the revised manuscript). To illustrate this point further, we have included data in Supplementary Fig. 1 showing that BC21 interferes with the β -catenin-TCF4 interaction in parallel to experiments showing that BC21 evicts AHCTF1 from the OSE in Fig 3D. It is important to note that we optimized the concentration of BC21 treatment in order to achieve a reduction in β -catenin-TCF4 interactions without affecting *MYC* transcription, as detailed in ref nr 7 of the revised manuscript.

3. GSM749690 is publicly available CTCF-ChIP-seq data of K562 from ENCODE, which is irrelevant to HCT116 from the current study. The author should provide ChIP data between WT and CBS-mutant to rule out the possibility of an off-target effect on a genome-wide scale.

Our response:

We apologize for the mis-annotation of the CTCF ChIP seq data. In fact, we used our own ChIP seq data generated from the WT HCT-116 cells presented in this study. However, the revised manuscript now includes a new set of CTCF ChIP-seq experiments including both new WT HCT-116 cells as well as the mutant D3 and E4 clones (new Fig. 1C). The new data has been uploaded to GEO (GSE182556) (see also the reporting summary).

4. Statistical analysis and justification are not provided at all in the method session, which is problematic for me to judge the data quality.

Our response:

The statistical analyses used well established tools, which were defined in our recent nature genetics report (ref nr 7 in the revised manuscript). Just in case, we have included a section in the Methods detailing our rationale for using either the two-tailed Student's t-test or the non-parametric KS test to examine statistical significance.

Reviewer #2 (Remarks to the Author):

Chachoua et al. aim to dissect the mechanism of « gene gating » as a process of MYC overexpression in colon cancer cells. “Gene gating” facilitates gene overexpression via their juxtaposition to the nuclear pore in order to rapidly export mRNAs in the cytoplasm. They showed that a CTCF binding site (CTCFBS) within an oncogenic super-enhancer (OSE) of MYC is involved in the recruitment of the nuclear-pore factor AHCTF1 in a WNT signaling pathway dependent manner in order to promote pathological nuclear export of MYC mRNAs. This study provides important insights into the gene gating process in human cells, and in pathological conditions. This could open new avenues in defining new therapeutic strategies. However, some questions should be addressed.

1) HCT-116 and HCEC cells could be briefly introduced in the first section of the results.

Our response:

This request has been implemented in the revised manuscript

2) Figure 1:

- Fig1A: the scheme is not clear. What are the orange and red boxes? Although the numbers below the scheme must be genomic coordinates, it is not clear what they represent. OSE and CCAT1 may be shown.

Our response:

The numbers below the scheme are indeed genomic coordinates. This has been spelled out in the revised legend of the Figure to avoid confusion. Although the *CCAT1* gene is a very small part of the OSE, its position within the OSE has been indicated in the new Fig. 1A. The orange boxes depicting the edited bases have been changed to grey to avoid confusion with the orange boxes within the OSE and *MYC* map, which identify enhancer regions. Just in case, we have further highlighted the meaning of the orange and grey boxes in the Results section.

- Fig1B: The authors use 2 mutant clones for OSE (D3 and E4). Are the 2 OSE alleles mutated? Do these 2 clones harbor the exact same mutations? Did the authors check for potential off-targets?

Our response:

We have sequenced the whole genomes for WT, D3 and E4 cells and checked for potential off-targets without finding any. Please refer to our more detailed response to Reviewer 1 on the same topic and a new Fig. 1B that visualizes the identical sequences at the *CCAT1*-specific CTCF binding site in the D3 and E4 cells.

- Fig1G: What are the mutant CTCFBS? D3 or E4 or else?

Our response:

This has now been clarified in the new Supplementary Fig. 1C., which shows data for both D3 and E4 cells

- The statistics are not clear. In the different panels, it is not clear whether they are calculated only of E4 clone or for both mutant clones combined.

Our response:

The presentation of all combinations of p values in each individual panel would run the risk of blurring the main messages. The lines under the p values identify the two sample sets that were compared at their endpoints. Just in case, these have been marked by vertical bars in the revised Figures. The availability of the Source data file makes it possible for anyone interested to assess the p values of data combinations not addressed in the manuscript. Nonetheless, we have added more p values in the images where we considered it would not blur the messages.

- It is not clear why there are 3 copies of MYC/FAM49B alleles in WT cells. What are exactly these WT cells? Do they thus represent the ideal control?

Our response:

The WT cells represent the parental HCT-116 colon cancer cell population. It is quite common that cancer cells are heterogenous and aneuploid to harbor extra copies of growth-promoting genes. To rule out if the representation of the OSE at the nuclear periphery was in any way allele-specific in the WT HCT-116 cells - potentially compromising the comparisons of the mutant D3 and E4 cells to WT HCT116 cells - we recalculated the data from Fig. 2m of our earlier nature genetics paper (ref nr 7 in the revised manuscript).

compromising the comparisons of the mutant D3 and E4 cells to WT HCT116 cells - we recalculated the data from Fig. 2m of our earlier nature genetics paper (ref nr 7 in the revised manuscript). The image to the left shows the number of OSE alleles at the periphery in individual cells to document that the simultaneous presence of 2 and 3 alleles at the periphery could be detected in as much as one third of the WT HCT-116 cell population examined.

We conclude that there is little or no evidence of any allele-specific difference in the ability of the OSE alleles to migrate to the nuclear periphery.

3) Figure 2:

- This is a detail but, although it seems clear that these experiments are performed in HCT-116 cancer cells, it may be mentioned.

Our response:

The HCT-116 cells have now been identified both in the abstract and in the introduction.

- Fig2A-B:

i) The ISPLA assay shows proximity between CTCF and AHCTF1 in absence or presence of BC21 inhibitor. A control in WT cells could have been added there.

Our response:

The control data, i.e. the WT HCT-116 cells, was included in the original manuscript (Figure 2A,B – now Figure 3J,K in the revised manuscript). We would like to highlight that the gating of *MYC* does not occur in normal, primary human colon epithelial cells (ref nr 7 in the revised manuscript).

ii) The ISPLA signals should be localized close to the nuclear periphery. Their distance to the nuclear periphery could have been measured and shown like in Fig2H.

Our response:

We have now included information with respect to the average distribution of the ISPLA signals in the Supplementary Fig. 2E.

According to published data, AHCTF1 distributes not only to the nuclear periphery, but also to some extent to the nucleoplasm⁶. Our data also demonstrates that CTCF-AHCTF1 are more proximal to each other primarily when relatively close to but not at the periphery. This is in line with our finding that AHCTF1 is required not only for the anchoring of the OSE to the pores but also for the OSE to travel from regions close to the periphery (about 0,7 μm) to positions at the periphery/pore. This discussion is included in the revised manuscript using a new model presented in the new Fig. 4.

iii) One question is the localization of the OSE in the nuclear space in WT and cancer cells? Are they, by default (in WT cells), localized in particular regions of the nucleus?

Our response:

Please note that the WT cells refer to the HCT-116 cancer cells (see also next point) that did not undergo CRISPR editing. This information was thus provided both in the Fig 3 (Fig. 5 in the revised manuscript) of the original manuscript and in our recent nature genetics report (ref nr 7 in the revised manuscript). We would like to highlight that the nature genetics report identified at least two discrete populations of OSE alleles with respect to their positions to the nuclear periphery in terms of patterns of enhancer-*MYC* interactions and *MYC* transcription (ref nr 7 in the revised manuscript). However, the simultaneous presence of up to three OSE alleles at the nuclear periphery suggests that these two OSE allele populations are dynamically interchangeable (ref nr 7 in the revised manuscript).

iv) Is there a difference in the amount of AHCTF1 proteins in cancer versus WT cells? If there is more AHCTF1 proteins in cancer cells, its association with CTCF may be aspecific and due to its overrepresentation. This control may be added in the study. In a similar way, is there a difference in the average number of nuclear pores in cancer versus WT cells?

Our response:

We thank the reviewer for bringing up this question. In the revised manuscript we have examined the expression levels of both protein and mRNAs for AHCTF1, NUP133 and CTCF in WT HCT-116 cells and the mutant D3 and E4 cell clones, and we found no discernible

difference (this data is now included in the Supplementary Fig. 2A,B). We would also like to emphasize that the gating principle that we have discovered does not exist for *MYC* in normal human colon epithelial cells (ref nr 7 in the revised manuscript). We have, moreover, not observed any measurable difference in the number of nuclear pores between WT and D3/E4 cells, as measured by immunofluorescence.

- Fig2C/D/F: co-immunoprecipitation CTCF-AHCTF1 is shown in cells with WT OSE, in absence or presence of BC21. A control in mutant-OSE cells may have been added, as for the ChIP experiment (Fig2E).

Our response:

We thank the reviewer for this comment. This data, showing no statistically significant difference in the level of CTCF-AHCTF1 interaction between WT HCT-116, D3 and E4 cells, has been added to Supplementary Fig. 2C.

- Fig2J: The authors describe the model they elaborated with the different experiments. i) They wrote that CTCFBS recruits AHCTF1. Is it really CTCFBS or CTCF bound to CTCFBS? The role of CTCF is not described in the model.

Our response:

We thank the reviewer for bringing up this important question. According to published data, however, AHCTF1 is not able to bind to specific DNA sequences including CTCFBSs. In addition, BC21 treatment evicted AHCTF1 from the OSE chromatin (ref nr 7 in the revised manuscript) to strongly suggest an indirect binding of AHCTF1 to the OSE – likely to β -catenin when complexed to TCF4. New data on TCF4 binding sites that flank the *CCAT1*-specific CTCFBS reinforces the notion that the AHCTF1-OSE binding is in part mediated by the β -catenin-TCF4 complex (see Supplementary Fig. 4A,B) and in part by the CTCFBS when occupied by CTCF. To further validate this conclusion, we have also performed ChIP analyses of AHCTF1 binding to the CTCFBS within the OSE in cells where we knocked down CTCF expression by siRNA. The results confirm that systemic reduction of CTCF expression leads to a reduction in AHCTF1 binding to the CTCFBS within the OSE, to similar levels observed in the D3/E4 cell clones (new Fig. 3E). Since both CTCF and β -catenin appears essential for efficient binding of AHCTF1 to the OSE, we propose that these factors synergize in promoting AHCTF1 presence at the *CCAT1*-specific CTCFBS (new Fig. 4)

ii) They argue that NUP133 and b-cat are indirectly recruited to the OSE. I may have missed a point but I am not sure that this indirect recruitment is formally shown.

Our response: To the best of our knowledge, neither NUP133 nor β -catenin are able to bind specific DNA sequences. For example, the current dogma argues that β -catenin binds to chromatin *via* TCF4/TCF712 binding sites. Thus, when treating HCT-116 cells with BC21, both β -catenin and NUP133 were evicted from the OSE in equal measures, while they remained complexed to each other (ref. nr 7 in the revised manuscript), indicating that these factors are complexed to each other when indirectly binding to the OSE. This

information has been further discussed in the revised manuscript (new Fig. 4, see also above).

iii) The authors showed, with the BC21 inhibitor, that the WNT signaling pathway is involved in the recruitment of AHCTF1 to the OSE. Is the WNT pathway involved in the recruitment of CTCF on the CTCFBS? In other words, did the authors perform a CTCF-ChIP in presence of BC21?

Our response: We thank the reviewer for bringing up this point. We would like to note, however, that it has long been known that CTCF binds to unmethylated DNA *via* its ZF domain with a very low K_d ⁷. In fact, this binding is sometimes stronger than the biotin-avidin interaction. Although it is also known that CTCF can bind indirectly to chromatin⁸, this feature involves chromatin regions devoid of the core CTCF binding sequence motive that we mutated at the OSE. Nonetheless, the requested experiment has been implemented with the result showing that BC21 has no effect on the ability of CTCF to occupy its binding site within the OSE. This data is now presented in the Supplementary Fig. 2D.

iv) As the WNT pathway is involved in the recruitment of AHCTF1, I am confused by the statement that CTCFBS indirectly recruits β -cat, which is part of the WNT pathway. Since AHCTF1 binds β -catenin much more efficiently than CTCF we argue that the β -catenin role is indirect.

Our response:

We apologize for not being clearer with this sentence, which has been reformulated extensively. In addition, the revised manuscript contains the new information that the CTCFBS region efficiently binds TCF4 (*via* motifs flanking the CTCFBS) as well as β -catenin in both WT and D3/E4 cells. Importantly, while treatment of WT HCT-116 cells with BC21 removes a significant portion of the AHCTF1 signal (ref nr 7 and this manuscript), AHCTF1 binding to the mutant CTCFBS is similarly reduced (Fig. 3G in the revised manuscript) despite unaltered TCF4 and β -catenin binding to the OSE (Supplementary Fig. 4B). Since BC21 also reduces the physical interaction between CTCF and AHCTF1 (new Fig. 3F), we propose that the juxtaposition of the CTCF and TCF4 binding sites stabilizes the presence of AHCTF1 on the OSE *via* CTCF and β -catenin. This is explained in some more detail in a new model (Fig. 4 in the revised manuscript) to highlight the indirect nature of the CTCF- β -catenin interaction.

4) Figure 3:

The authors explored the nuclear localization of the MYC alleles in HCT-116 cells with WT versus mutated CTCFBS. They found that the proximity of both OSE and MYC to the nuclear periphery is reduced in mutated-CTCFBS cells.

- Are these differences in proximity to the nuclear periphery statistically significant?

Our response: We thank the reviewer for pointing out this issue. The new Figure 5 (previous Fig. 3) illustrating this data shows that the difference in the accumulated

distribution of the OSE alleles between the WT HCT-116 and D3/E4 clones is statistically significant within one micrometer from the periphery. We note that also the *MYC* region is unable to efficiently reach the periphery in the D3/E4 clones (displaying statistically significant difference from WT HCT-116) – thus reinforcing that its presence at nuclear pores is dependent on an intact CTCFBS positioned within the *CCAT1* gene at the OSE in WT cells.

- It could have been interesting to analyze the distance to the nuclear periphery of OSE and MYC in WT colon cells, as well as in HCT-116 cells treated with BC21 and/or siAHCTF1 in order to control the role of the WNT-pathway and AHCTF1 in the localization of MYC and OSE.

Our response: These experiments have in part been published (ref nr 7 in the revised manuscript) and were in part included in Fig. 2 and 3 in the original manuscript (now Fig. 3G-I and Fig. 5A,B in the revised manuscript).

5) Figure 3/4:

The authors further showed that the mutations induced in CTCFBS, although important for the recruitment to the nuclear pore, is not involved in OSE-MYC interactions. Moreover, expression of the *CCAT1* eRNA close to the periphery does not seem to be involved in the OSE-MYC interactions. Do the authors have any hypothesis on the mechanism of OSE-MYC interactions?

Our response: We thank the reviewer for bringing up this question. Since the OSE-MYC interactions occur throughout the entire OSE, also in regions devoid of any other CTCFBS (new Fig. 5F), other factors must be involved in mediating OSE-MYC interactions. A likely candidate is the Mediator complex that decorates the entire OSE⁹ and which has been shown to direct enhancer-gene communications¹⁰. This discussion is now included in the revised manuscript.

6)

Discussion:

- The authors hypothesize that the recruitment of AHCTF1 to the OSE occurs at perinuclear positions. Would it be possible to perform combined immunofluorescence of AHCTF1 with DNA FISH for OSE in order to validate this hypothesis?

Our response:

Unfortunately, such an experiment would not be informative. The reason is that the resolution of the light microscope (0,3 x 0,3 x 0,5 micrometers in the X x Y x Z dimensions) is limited by the properties of the fluorophores, which thus do not offer sufficiently good resolution to enable us to make this connection. We are currently trying to use the ChrISP technique, which has a much better resolution (<160Å in all three dimensions), to explore the timing of factor recruitment to the OSE in relation to the nuclear architecture. This far, the epitopes of both AHCTF1 and CTCF have not - in contrast to NUP133 (ref nr 7 in the revised manuscript) - survived the DNA FISH hybridization conditions, which is a necessary requirement for an efficient ChrISP assay.

- Do the authors have any hypothesis on which of the WNT signaling pathway component could be directly involved in this process?

Our response:

The results clearly point to an involvement of β -catenin. This notion is supported by the significant effects of BC21 (that specifically acts by disrupting the β -catenin-TCF4 complex and removes β -catenin from the OSE chromatin), which leads to the reduced binding of AHCTF1 to the *CCAT1*-specific CTCFBS (new Fig. 3D,F). Moreover, BC21 reduces the level of physical interaction between CTCF and AHCTF1 (new Fig. 3C in the revised manuscript). Based on this and other data we discuss in a new Fig. 4 the possibility that β -catenin stabilizes the CTCF-AHCTF1 complex.

Reviewer #3 (Remarks to the Author):

Summary

Chachoua et al. describe a mechanism where a single CTCF binding site within an oncogenic super enhancer (OSE) facilitates the trafficking of the MYC gene to the nuclear periphery to increase MYC mRNA export following WNT signaling, specifically in the HCT116 colon cancer cell line, but not in normal colon epithelial cells. To study this mechanism, the authors mutated the CTCF binding site within the OSE and studied how the mutation affects MYC gene expression, mRNA export, ability to recruit AHCTF1 (nucleoporin that's essential for gene gating), and association with the nuclear periphery. The CTCFBS is in an eRNA gene, *CCAT1*, which mechanistically they show is important to tether MYC to the nuclear pore for efficient gene gating.

Major Concerns

1) All of the experiments are carried out in a single colon cancer cell line. Demonstrating that this is relevant to colon cancer and/or other cancer cell lines is important to assess the significance of the report.

Our response:

This is an important point although we consider this to be a topic of follow-up stories. We have thus documented the trafficking of the breast cancer OSE to the nuclear periphery as well as the facilitated nuclear export of *MYC* mRNAs also in a breast cancer cell line. Applying the ChrISP technique to thin sections of breast tumors, we have been able to show that the breast cancer OSE distal to *MYC* is anchored to the

Proximity between the breast cancer-specific OSE and NUP133 in a blood vessel of thin section of an ER/PR-positive breast cancer. Left: ChrISP signals (green); Right: OSE DNA FISH signals (yellow). The ChrISP signal is always at the nuclear periphery (top insert).

nuclear pore in cancer cells and endothelial cells, but not in the stroma of luminal breast cancer (see examples to below). However, these data are, in comparison with the current

ChrISP analysis of the proximity between the breast cancer-specific OSE and NUP133 in a thin breast cancer section. Left: NUP133 staining. Middle: a magnified view of green ChrISP signals (arrows). Left: merged NUP133 (red), ChrISP (green), OSE FISH signal (white) and DAPI staining to highlight proximities between NUP133 at the nuclear pore and the OSE specifically in cancer.

detailed focus on HCT-116, too fragmentary to motivate their inclusion. Moreover, such data would impede the logical flow of the manuscript as well as posing significant logistic problems of fitting the manuscript within the journal space limitations.

Nonetheless, these preliminary results have been mentioned in the discussion of the revised manuscript to make the point that the observations using the HCT-116 cells do not depend on a unique model system.

2) The claim that a single CTCFBS in the OSE is the relevant controller for MYC gene gating should include controls mutating other CTCF binding sites in/relevant to the MYC gene to rule out general disruption of chromatin architecture resulting in unrelated loss of MYC gene gating.

Our response: We considered this issue when starting the entire project. However, since the profound effect on *MYC* gating could be observed already with our first chosen target for editing, i.e. the CTCFBS within the *CCAT1* gene, which reduced the gating of *MYC* without impeding OSE-*MYC* interactions, there was little point in pursuing other CTCF binding sites. There are additional reasons for this decision: first, it would not be a trivial task to limit the region of interest. That is, should the selection be focused on the immediate vicinity or cover several hundred kbs, i.e. all the way to the *MYC* gene? Second, even if the mutation of another CTCFBS would show the same net effect, the risk that this would be due to impaired OSE-*MYC* interactions rather than involving the direct targeting of the OSE to the nuclear pore *via* AHCTF1 must be both considerable and much less interesting. Third and most important, including other targets for editing in the manuscript would seriously impede the in-depth analyses required for robust conclusions due to journal space limitations.

As to the chromatin architecture, our Nodewalk analyses have not revealed any disruptive effect of the mutated CTCFBS in the region spanning the OSE and *MYC*. Indeed, other reports including the Spencer (2020) report, have reported generally modest or no effects on higher order architectures over large domains in cells with mutated or deleted CTCF binding sites. Of course, this does not rule out that a smaller subset of CTCF binding sites, usually with inverse orientations, do influence TAD structures, for example. The point here is that this feature does not apply to the *CCAT1*-specific CTCFBS focused on in this report (see Fig. 5F).

3) For relevance to Wnt signaling, other methods in addition to the BC21 inhibitor should be used.

Our response: We do not think that this is applicable to our model system. The reason is that β -catenin is mutated in HCT116 cells¹¹. This was thus the reason to focus on an as late stage of the WNT signaling pathway as possible. Moreover, using inhibitors of the pathway upstream of β -catenin would complicate our ambition to discriminate between canonical (i.e. β -catenin) and non-canonical (i.e. not involving β -catenin) WNT signaling.

4) In addition to the co-culture experiment followed by qPCR to show a growth advantage in figure 1, a proliferation assay in the mutant cells would bolster the claim that the CTCF-binding site confers an increase in growth advantage. In addition, an analysis of the mutant having a reduced ability to respond to WNT signaling would strengthen the report.

Our response:

It is not clear to us what a proliferation assay of the mutant cells would add that was not already measured by the co-culture experiments. These experiments thus have a superior in-built control by directly comparing the proliferative phenotypes within two different cell populations. We do appreciate the criticism that the effect of BC21 should be included. The revised manuscript now contains this additional data, showing that BC21 reduces the proliferative advantage of WT cells 6-fold in comparison to the D3/E4 clones under the conditions examined.

5) The co-IP experiment in figure 2 shows the interaction between CTCF, AHCTF1, and Nup133, however, to be more confident of the location of this interaction, an In-Situ PLA experiment would be a good addition.

Our response:

The CTCF-AHCTF1 ISPLA data was included in the original ms (Fig 2A,B). Moreover, we have earlier shown that NUP133 becomes associated with the OSE when proximal to the nuclear periphery (ref nr 7 in the revised manuscript). Please note, however, that the ISPLAs have a resolution of $<160\text{\AA}$ to document primarily the potential for interactions. Direct analyses of physical interactions, typically within a few \AA , requires either co-immunoprecipitation analyses or cross-linking within monomeric formaldehyde, as in ChIP analyses. Nonetheless, we have now included CTCF-NUP133 ISPLA data in a new Supplementary Fig. 3. Since NUP133 is part of the pre-nucleopore complex that indirectly binds to chromatin *via* AHCTF1, this data agrees well to the CTCF-AHCTF1 ISPLA signals.

6) For the 3D FISH experiment in figure 2 where AHCTF1 knockdown resulted in the OSE further from the nuclear periphery, statistical analysis of the 3D FISH experiment should be included.

Our response:

We thank the reviewer for pointing out this issue. The revised manuscript now includes statistical analyses of this data.

7) Figure 3 needs statistical significance determinations.

Our response:

The revised manuscript now includes statistical analyses of this data.

8) In figure 3, the E4 mutant showed that MYC's connection to the OSE was unaffected while WT had a significant change in association. Showing that the D3 mutant was unaffected as well would add confidence to the claim.

Our response:

The revised manuscript now includes D3 data and associated statistical analyses.

9) Addition of the D3 mutant to figure 4 experiments would add strength. E4 is actually a test here as the effect of the CTCF mutation is stronger than the effects of D3.

Our response:

The revised manuscript now includes D3 data with associated statistical analyses. As can be seen from the new image in Fig. 6, the data for both D3 and E4 clones are very similar.

10) The strong claim that this study was the first genetic evidence of gating/trafficking phenomenon in human cells is not supported by current literature. The inclusion of Kadota et al., nature communications (2020) outlines an epigenetic memory at the nuclear pore which includes the gene gating mechanism.

Our response:

The report by Kadota addressed epigenetic features only. We respectfully disagree with the Reviewer that epigenetic data equals genetic data - in our case a targeted mutation of a DNA sequence.

Minor Concerns

1) In figure 3B, the y-axis says the measurement is a percentage but has units of "(µm)".

Our response:

We thank the Reviewer for pointing out this error, which has been corrected.

2) More clear experimental descriptions would improve the read of the paper.

Our response:

We have tried to improve the experimental description in the revised manuscript.

- This study is interesting and opens new avenues in the understanding of gene regulation at the post-transcriptional level, but also of oncogene regulation, cancer initiation and progression. However, this was all done in one cell line. How much these results can be applied to patients?"

Our response:

This issue is addressed above. We agree with the Reviewer that this is an important follow-up point. Indeed, we have already observed that one feature of the MYC gating

process (namely, the juxtaposition of the OSE to the nuclear pore specifically in cancer cells and endothelial cells within the tumor) can be observed in thin sections of a luminal breast cancer (see above). However, this is a separate story that requires work on breast cancer explants to score for nuclear export rates *ex vivo* as well as a larger set of patient material to stratify patterns. It is most unfortunate that the current pandemic has impeded the implementation of new sample collection within this project.

References

- 1 Zhao, Z. *et al.* Circular chromosome conformation capture (4C) uncovers extensive networks of epigenetically regulated intra- and interchromosomal interactions. *Nat Genet* **38**, 1341-1347, doi:10.1038/ng1891 (2006).
- 2 Sandhu, K. S. *et al.* Nonallelic transvection of multiple imprinted loci is organized by the H19 imprinting control region during germline development. *Genes Dev* **23**, 2598-2603, doi:10.1101/gad.552109 (2009).
- 3 Kim, J. & Kim, J. YY1's longer DNA-binding motifs. *Genomics* **93**, 152-158, doi:10.1016/j.ygeno.2008.09.013 (2009).
- 4 <https://sunlab.cpy.cuhk.edu.hk/YY1TargetDB/>.
- 5 Zlatanova, J. & Caiafa, P. CTCF and its protein partners: divide and rule? *J Cell Sci* **122**, 1275-1284, doi:10.1242/jcs.039990 (2009).
- 6 Gillespie, P. J., Khoudoli, G. A., Stewart, G., Swedlow, J. R. & Blow, J. J. ELYS/MEL-28 chromatin association coordinates nuclear pore complex assembly and replication licensing. *Curr Biol* **17**, 1657-1662, doi:10.1016/j.cub.2007.08.041 (2007).
- 7 Ohlsson, R., Renkawitz, R. & Lobanenkov, V. CTCF is a uniquely versatile transcription regulator linked to epigenetics and disease. *Trends Genet* **17**, 520-527 (2001).
- 8 Mukhopadhyay, R. *et al.* The binding sites for the chromatin insulator protein CTCF map to DNA methylation-free domains genome-wide. *Genome Res* **14**, 1594-1602, doi:10.1101/gr.2408304 (2004).
- 9 Hnisz, D. *et al.* Super-enhancers in the control of cell identity and disease. *Cell* **155**, 934-947, doi:10.1016/j.cell.2013.09.053 (2013).
- 10 Malik, S. & Roeder, R. G. The metazoan Mediator co-activator complex as an integrative hub for transcriptional regulation. *Nat Rev Genet* **11**, 761-772, doi:10.1038/nrg2901 (2010).
- 11 Ilyas, M., Tomlinson, I. P., Rowan, A., Pignatelli, M. & Bodmer, W. F. Beta-catenin mutations in cell lines established from human colorectal cancers. *Proc Natl Acad Sci U S A* **94**, 10330-10334, doi:10.1073/pnas.94.19.10330 (1997).

REVIEWER COMMENTS

Reviewer #1 (Remarks to the Author):

The authors have addressed most of my previous comments appropriately. I have one more question regarding ChIP-seq tracks in figure 1D. The y-axis scale is set up differently among WT and mutant clones (WT: 0-6,23; D3: 0-7.54 and D4: 0-12. Although it seems the edited CBS is the only one peak showing binding affinity change as indicated by the arrow, after normalization to the same scale, the left binding affinity of the left CTCF-peak seems to be increased for 2-fold. The authors should keep all scales as the same range and also provide explanation for the binding affinity change in D4. Also, the axis labeling in supplementary figure 1b is confusing to me. The scales for different tracks embedded in the figure are different but were re-labeled the same as 0-5.5. Any explanation?

Reviewer #2 (Remarks to the Author):

The manuscript has been extensively improved by the authors. This work gives a first extensive description of the gating process in one human colon cancer cell line. This is of significant interest for the fields of genetics, post-transcriptional regulation, oncogene regulation, cancer.

There are still few questions or comments:

1) The authors have improved the statistical analyses (Fig 1E, Fig 3G, Fig 5D...). Although I agree with the authors that adding all the p-values in the figures may have blurred the message, the authors could have provided all the statistics in Supplementary tables for example.

2) I am still confused by the explanations of the authors about the fact that there are 3 copies of MYC/FAM49B in the WT HCT-116 cells but only 2 copies in both mutant HCT-116 clones. In particular, I am wondering whether the transcription or cytoplasmic mRNA ratios calculated over control genes can be compared if the copy numbers of MYC/FAM49B are not similar in the WT versus mutant cells, but the control genes are (ie b-actin and TBP).

- In their response the authors state that the parental WT HCT-116 colon cancer cells are a heterogeneous population. In this case, it means that this population is a mix of cells with different copy numbers of these genes. This cell line is described as nearly diploid, with some polyploidy occurring at around 7% (from ATCC website). As it's a heterogeneous population, can it formally be stated that these WT cells harbor 3 copies of the MYC/FAM49B genes?

- If I understand correctly this means that both mutant clones come from one diploid cell out of this heterogeneous population. The authors state in the manuscript that MYC/FAM49B transcription per allele is higher in the mutants than in the parental WT cells. How could it be explained? The transcription rate in Fig 2B is calculated as a ratio with b-actin control. Could it be that there are also extra copies of the b-actin gene in the WT heterogeneous cell population? In a similar way, Fig 2C presents a ratio of MYC/FAM49B cytoplasmic mRNA over control TBP mRNA (Figures 6E and F present also ratios between CCAT1 and TBP). How many copies of TBP are in the WT cells?

- I don't understand the sentence "Although this observation is not statistically significant, it reinforces the notion that the OSE does not influence MYC and FAM49B transcription per se". Which "observation" is not statistically significant? How can they state that OSE does not influence transcription per se?

3) The sentence "Indeed, in situ proximity analyses (ISPLA) showed that the highest potential for interactions between CTCF and AHCTF1 spanned a region 1-2 micrometers distal to the nuclear periphery, similar to the CTCT-NUP133 ISPLA signals" suggests that CTCF-NUP133 ISPLA signals are mostly located in the same nuclear area than CTCF-AHCTF1. However Supplementary Fig 3 does not show any of this measurement.

4) Fig 5C/D: I would have appreciated to see one example of OSE/MYC DNA FISH image.

5) Fig 5: The authors state "Although such data seem to indicate that CTCF directly facilitates communications between the OSE and MYC, this is likely not the case". I am wondering whether "such data" refers to the previous section showing that in WT cells OSE and MYC tend to be at the same distance from the nuclear periphery. If it is the case, I am not sure that showing that 2 loci are at the same distance from one region of the nucleus means that they interact together.

6) It is not clear how RNA/DNA FISH were performed. Were they performed simultaneously or sequentially (I went back to the ref 7 but could not find a clear answer either).

7) In their response to reviewers, the authors mention that the HCT-116 cells are mutated for b-catenin. What is this mutation? Would it have implications for the whole gating process?

Responses to the Reviewers' comments

We thank the Reviewers for carefully scrutinizing our manuscript. Our detailed responses are listed below with changes in the main manuscript marked in green.

Reviewer #1 (Remarks to the Author):

The authors have addressed most of my previous comments appropriately. I have one more question regarding ChIP-seq tracks in figure 1D. The y-axis scale is set up differently among WT and mutant clones (WT: 0-6,23; D3: 0-7.54 and D4: 0-12. Although it seems the edited CBS is the only one peak showing binding affinity change as indicated by the arrow, after normalization to the same scale, the left binding affinity of the left CTCF-peak seems to be increased for 2-fold. The authors should keep all scales as the same range and also provide explanation for the binding affinity change in D4. Also, the axis labeling in supplementary figure 1b is confusing to me. The scales for different tracks embedded in the figure are different but were re-labeled the same as 0-5.5. Any explanation?

Our response:

The aim of the CTCF ChIP-seq experiments in the WT HCT116 cells and mutant D3 and E4 clones was to rule out any potential off-target mutation of the CRISPR-based genome editing at other CTCF binding sites with potential sequence similarity to the targeted CTCF site within the OSE. The results thus confirm that the CRISPR approach did not interfere with CTCF binding to sites located outside the targeted OSE. Importantly, using quantitative QPCR analyses, we show that the CTCF occupancy at an internal control, the well-defined CTCF-bound *H19* imprinting control region, is very similar between the different ChIP preparations, indicating a reproducibility in the procedure. However, ChIP-seq results are to our knowledge at best semi-quantitative when comparing different samples due to potential variations in the sonication and the library preparation steps that include size selections, for example. As ChIP-seq analysis can much more reliably quantitate difference in factor binding within the same sample than between different samples, we have presented the different absolute values of the samples without normalization to the same scale.

We apologize for the erroneous scales on the ChIP seq data in the supplementary Fig. 1b although we note that the correct version was included in the file comparing the original and revised manuscripts. This mistake has now been corrected.

Reviewer #2 (Remarks to the Author):

The manuscript has been extensively improved by the authors. This work gives a first extensive description of the gating process in one human colon cancer cell line. This is of significant interest for the fields of genetics, post-transcriptional regulation, oncogene regulation, cancer. There are still few questions or comments:

1) The authors have improved the statistical analyses (Fig 1E, Fig 3G, Fig 5D...). Although I agree

with the authors that adding all the p-values in the figures may have blurred the message, the authors could have provided all the statistics in Supplementary tables for example.

Our response: We are not aware of any publication in nature journals, in which all possible combinations of *P* values have been included in a Source data file. However, as readers will have access to the Source data file, the data is readily available for the examination of any *P* value of interest. We want to emphasize that all relevant comparisons have in our opinion been made to support all claims of the manuscript, allowing us to make statistically robust conclusions as to the observed differences between WT and edited cells.

2) I am still confused by the explanations of the authors about the fact that there are 3 copies of MYC/FAM49B in the WT HCT-116 cells but only 2 copies in both mutant HCT-116 clones. In particular, I am wondering whether the transcription or cytoplasmic mRNA ratios calculated over control genes can be compared if the copy numbers of MYC/FAM49B are not similar in the WT versus mutant cells, but the control genes are (ie b-actin and TBP).

Our response: The equations calculating the nuclear export rate of mRNAs and the transcription level per cell are independent of the number of alleles per cell, relying entirely on the sum of newly synthesized nuclear versus cytoplasmic transcripts.

We would, moreover, like to highlight that the level of control β -actin and TBP mRNAs correctly estimate the input number of cells both in the case of WT HCT-116 cells and the mutant clones. We have included this data in the second revision of our manuscript in a new Supplementary Fig. 2. Apart from precisely reflecting the input number of cells, the data show that there is no significant difference in the level of expression between β -actin and TBP mRNAs per cell in WT and mutant cells, reinforcing the conclusions made in the original manuscript.

- In their response the authors state that the parental WT HCT-116 colon cancer cells are a heterogeneous population. In this case, it means that this population is a mix of cells with different copy numbers of these genes. This cell line is described as nearly diploid, with some polyploidy occurring at around 7% (from ATCC website). As it's a heterogeneous population, can it formally be stated that these WT cells harbor 3 copies of the MYC/FAM49B genes?

Our response: Additional copies of growth promoting genes, such as *MYC*, are expected to be continuously selected for during *in vitro* propagation under constant culturing conditions if there is an initial imbalance. In our HCT116 cells that have undergone STR analyses, the percentage of cells with three *MYC* copies is approximately 93% based on DNA FISH analyses.

- If I understand correctly this means that both mutant clones come from one diploid cell out of this heterogeneous population. The authors state in the manuscript that MYC/FAM49B transcription per allele is higher in the mutants than in the parental WT cells. How could it be explained? The transcription rate in Fig 2B is calculated as a ratio with b-actin control. Could it be that there are also extra copies of the b-actin gene in the WT heterogeneous cell population? In a similar way, Fig 2C presents a ratio of MYC/FAM49B cytoplasmic mRNA over control TBP mRNA

(Figures 6E and F present also ratios between CCAT1 and TBP). How many copies of TBP are in the WT cells?

Our response: As stated above, the key here is the absolute level of expression per cell – information that is now included in a new Supplementary Fig. 2, as elaborated above. Since the transcriptional levels of *MYC* in D3 and E4 cells were statistically insignificant from WT cells, it seemed clear that *MYC* transcription per allele must be higher in the edited cells than in the WT cells.

- I don't understand the sentence "Although this observation is not statistically significant, it reinforces the notion that the OSE does not influence *MYC* and *FAM49B* transcription per se". Which "observation" is not statistically significant? How can they state that OSE does not influence transcription per se?

Our response: "This observation" relates to the last piece of information in the preceding sentence that deals with the estimated *MYC/FAM49B* transcription per allele. This deduction was derived from the transcription rates, which are very similar between the WT and edited cell clones despite that the edited cells contain two alleles while the WT cells contain primarily three alleles.

The sentence dealing with the "potential role of the OSE in enhancing *MYC* transcription" is now more precisely formulated. We thus argue that it is the OSE-CTCFBS (and not the entire OSE) that appears to have no function in regulating *MYC* transcription. This error has been corrected in the revised manuscript.

3) The sentence "Indeed, in situ proximity analyses (ISPLA) showed that the highest potential for interactions between CTCF and AHCTF1 spanned a region 1-2 micrometers distal to the nuclear periphery, similar to the CTCT-NUP133 ISPLA signals" suggests that CTCF-NUP133 ISPLA signals are mostly located in the same nuclear area than CTCF-AHCTF1. However Supplementary Fig 3 does not show any of this measurement.

Our response: This information is included in the new Supplementary Fig 4 (previous Supplementary Fig. 3)

4) Fig 5C/D: I would have appreciated to see one example of OSE/*MYC* DNA FISH image.

Our response: We did not include DNA FISH data generated by the OSE/*MYC* probes, as these have already been published in our nature genetics paper (Fig. 2f in ref nr 7). We know from previous experience that journals do not want to publish data/images that have been published earlier. m

5) Fig 5: The authors state "Although such data seem to indicate that CTCF directly facilitates communications between the OSE and *MYC*, this is likely not the case". I am wondering whether "such data" refers to the previous section showing that in WT cells OSE and *MYC* tend to be at the same distance from the nuclear periphery. If it is the case, I am not sure that showing that 2 loci are at the same distance from one region of the nucleus means that they interact together.

Our response: The “c” value indeed shows that in WT cells the OSE and *MYC* alleles tend to be at similar distance from the periphery, increasing their *potential* for interaction as the volume they sample decreases. The OSE-*MYC* proximity has thus been quantitated by ChrISP assay in our previous publication (Fig. x in ref nr 7) and was shown to be highest at the nuclear periphery of WT HCT-116 cells. In order to determine the frequency of *direct* interactions, we thus performed the Nodewalk (high sensitivity 3C) assay to complement the DNA FISH data.

6) It is not clear how RNA/DNA FISH were performed. Were they performed simultaneously or sequentially (I went back to the ref 7 but could not find a clear answer either).

Our response: To be able to localize the primary transcript in relation to its gene, all the RNA analyses have been followed by DNA FISH analyses, which were performed sequentially. The sequential nature of the RNA/DNA FISH analyses was visualized and described for Figure 2J in the legend as well as in the Results section of our nature genetics paper (ref nr 7 in the revised manuscript). We have included this information also in the legend of Figure 6 in the second revision of the manuscript.

7) In their response to reviewers, the authors mention that the HCT-116 cells are mutated for β -catenin. What is this mutation? Would it have implications for the whole gating process?

Our response: In the original publication referred to in the manuscript, it was concluded that the mutation of β -catenin enabled its escape from its cytoplasmic degradation pathway thereby promoting its distribution to the nucleus. Indeed, β -catenin shows a nuclear staining pattern in HCT116 cells, whereas it is almost exclusively distributed to the cell membrane in normal colon epithelial cells (Supplementary data in ref nr 7). For β -catenin to promote the gating of *MYC*, it very likely has to be present in the nuclear compartment. Moreover, given the activating nature of the mutations, inhibition of the WNT pathway can only be achieved in HCT116 cells by inhibitors that target events downstream of the mutant β -catenin, such as BC21 that is used in the manuscript.